# Scalable Option Learning in High-Throughput Environments

**Mikael Henaff** [1]   **Scott Fujimoto** [1]   **Michael Matthews** [1,2]   **Michael Rabbat** [1]

## Abstract

Hierarchical reinforcement learning (RL) has the potential to enable effective decision-making over long timescales. Existing approaches, while promising, have yet to realize the benefits of large-scale training. In this work, we identify and solve several key challenges in scaling online hierarchical RL to high-throughput environments. We propose Scalable Option Learning (SOL), a highly scalable hierarchical policy gradient algorithm which achieves a ~35× higher throughput compared to existing hierarchical methods. To demonstrate SOL's performance and scalability, we train hierarchical agents using 30 billion frames of experience on the complex game of NetHack, significantly surpassing flat agents and demonstrating positive scaling trends. We also validate SOL on MiniHack and Mujoco environments, showcasing its general applicability. Code is available at: https://github.com/facebookresearch/sol.

## 1. Introduction

Training agents to effectively solve decision-making tasks spanning long timescales is a fundamental challenge in reinforcement learning (RL) and control. This problem is difficult because the optimization landscape at the lowest level of sensorimotor control is often hard to optimize, due to sparsity of rewards or local minima. Consider a human whose goal is to go from New York to Paris. When viewed as an RL problem, where actions consist of joint movements and the cost is the distance to Paris, gradient information is often uninformative or misleading. The optimization landscape is rife with local minima, such as the agent becoming stuck in the easternmost corner of a room; and globally optimal trajectories, such as taking the subway to the airport, are likely to encounter many local increases in the cost function, making them difficult to discover.

Hierarchy presents itself as a natural approach to address this challenge (Sutton et al., 1999). At higher levels of the hierarchy, actions span longer timescales and are thus fewer, making for shorter decision-making tasks. Meanwhile, lower levels of the hierarchy aim to solve sub-tasks determined by the higher levels, which are also shorter and thus easier to optimize.

A significant body of work has explored ways to incorporate hierarchy into RL algorithms, through options (Sutton et al., 1999; Precup & Sutton, 2000; Bacon et al., 2017), feudal RL (Dayan & Hinton, 1992; Vezhnevets et al., 2017), and other manager-worker architectures (Nachum et al., 2018; Gürtler et al., 2021; Li et al., 2020; Levy et al., 2017). These methods have shown promising benefits of hierarchy over flat policies and laid important conceptual foundations. Nevertheless, by modern AI standards, they have remained in the relatively small data regime. Whereas flat RL agents and computer vision models are routinely trained on billions of samples (Radford et al., 2021; Kirillov et al., 2023; Ravi et al., 2025; Espeholt et al., 2018; Petrenko et al., 2020; Wijmans et al., 2019; Matthews et al., 2024) and language models on trillions of tokens (Brown et al., 2020; OpenAI, 2024; Kaplan et al., 2020; Touvron et al., 2023; Dubey et al., 2024; Team, 2024), existing hierarchical agents are typically trained on millions of samples only—several orders of magnitude less data. Therefore, hierarchical RL has yet to realize the benefits of large-scale training, which has driven progress in many other areas of machine learning.

In this work, we take a step towards enabling hierarchical RL at scale and present Scalable Option Learning (SOL). On the algorithmic side, SOL includes i) an objective which bootstraps option value functions upon option switches, allowing option policies to optimize independent rewards despite being trained with mixed behavior policies; and ii) a mechanism for automatically adapting option execution lengths based on the observation. On the systems side, we identify and solve several challenges which prevent straightforward scaling of hierarchical agents via GPU parallelization, enabling us to train with ~35-580× faster throughput compared to existing hierarchical RL algorithms. We apply SOL to the complex, open-ended NetHack Learning Environment (NLE) (Küttler et al., 2020) and train hier-

[1]Meta Superintelligence Labs [2]University of Oxford. Correspondence to: Mikael Henaff <mikael314@gmail.com>.

archical agents for 30 billion steps, significantly surpassing flat agents and demonstrating promising scaling trends. We additionally evaluate our algorithm on simpler Mini-Hack (Samvelyan et al., 2021) and PointMaze environments (de Lazcano et al., 2024), showcasing its general applicability.

## 2. Background and Problem Setting

### 2.1. Markov Decision Processes

We consider a standard Markov decision process (MDP) (Sutton & Barto, 2018) defined by a tuple $(\mathcal{S}, \mathcal{A}, P_0, P, R, \gamma)$ where $\mathcal{S}$ is the state space, $\mathcal{A}$ is the action space, $P_0$ is the initial state distribution, $P$ is the transition function, $R$ is the reward and $\gamma < 1$ is a discount factor. At the beginning of each episode, a state $s_0$ is sampled from $P_0$. At each time step $t \geq 0$, the agent takes an action $a_t$ conditioned on $s_t$, which causes the environment to transition to a new state $s_{t+1} \sim P(\cdot|s_t, a_t)$ and a reward $r_t = R(s_t, a_t)$ to be given to the agent. The goal is to learn a policy $\pi : \mathcal{S} \rightarrow \Delta(\mathcal{A})$ which maximizes the sum of discounted returns $\mathbb{E}_\pi[\sum_{t=0}^{\infty} \gamma^t r_t]$. We adopt the fully-observed MDP framework to simplify notation, however, states can be replaced by observation histories without loss of generality and our experiments include partially-observed environments.

### 2.2. Options

The options paradigm (Sutton et al., 1999; Precup & Sutton, 2000) provides a framework for decision-making at different levels of temporal abstraction. Each option $\omega \in \Omega$ represents a temporally extended behavior, and is defined by a tuple $(\pi_\omega, \mathcal{I}_\omega, \beta_\omega)$. Here $\pi_\omega : \mathcal{S} \rightarrow \Delta(\mathcal{A})$ is the option policy defining the agent's behavior while the option is being executed, $\mathcal{I}_\omega \subseteq \mathcal{S}$ is the initiation set of states where the option can be started, and $\beta_\omega : \mathcal{S} \rightarrow \{0, 1\}$ is the termination function indicating when the option should be ended. In addition to the option policies $\pi_\omega$, there is a controller policy $\pi_\Omega : \mathcal{S} \rightarrow \Delta(\Omega)$ which determines which option to execute next in a given state. Note that $\Omega$ could be infinite and continuously parameterized, in which case $\pi_\Omega$ has a continuous action space.

We adopt the call-and-return paradigm (Bacon et al., 2017; Klissarov & Precup, 2021) where options are sequentially executed. First, an option $\omega \sim \pi_\Omega(\cdot|s_0)$ is sampled. The agent then executes actions $a_t \sim \pi_\omega(\cdot|s_t)$ until $\beta_\omega(s_t) = 1$, after which a new option $\omega' \sim \pi_\Omega(\cdot|s_t)$ is sampled. The agent then executes actions sampled from $\pi_{\omega'}$ until the termination function $\beta_{\omega'}$ is activated, and the process continues.

A central question is how to define or learn the options $(\pi_\omega, \beta_\omega, \mathcal{I}_\omega)_{\omega \in \Omega}$ as well as the controller policy $\pi_\Omega$. Previously studied settings include: i) predefining option policies $\pi_\omega$ and subsequently learning $\pi_\Omega$ (Sutton et al., 1999), ii) pre-specifying or incrementally adding goal states and learning option policies using distances to goals as rewards (McGovern & Barto, 2001; Stolle & Precup, 2002; Menache et al., 2002), iii) jointly learning both option policies and the controller policy end-to-end using the task reward alone (Bacon et al., 2017; Li et al., 2020; Nachum et al., 2018; Klissarov et al., 2017; Klissarov & Precup, 2021).

In this work, for most of our experiments we assume access to a set of intrinsic reward functions $\{R_\omega\}_{\omega \in \Omega}$, and focus on jointly learning corresponding option policies $\pi_\omega$ and the controller $\pi_\Omega$ in a scalable manner. This assumes more prior knowledge than purely end-to-end options approaches, however, we also show in Section 5.3 that SOL can be paired with methods for automatic reward synthesis which do not assume any prior knowledge. Our scalable algorithm is agnostic to how option rewards are defined, and can be used whether they are handcoded or learned.

### 2.3. High-throughput RL

Several algorithms and libraries have been proposed to efficiently train RL agents on billions of samples, such as IMPALA (Espeholt et al., 2018), MooLib (Mella et al., 2022), RLLib (Liang et al., 2018; Wu et al., 2021), Sample Factory (Petrenko et al., 2020) and Pufferlib (Suarez, 2024). A common feature is asynchronous data collection paired with parallelized policy updates. A set of actors, typically operating across multiple CPU cores, collect experience from many parallel environment instances. Concurrently, a learner process receives batches of experience and updates the policy using parallelized GPU operations. The objective optimized is typically the policy gradient objective (Williams, 1992) with modifications such as PPO's trust region constraints (Schulman et al., 2017) and/or IMPALA's V-trace off-policy correction. Additional systems-level optimizations such as double-buffered sampling can also be included (Petrenko et al., 2020). However, all the above high-throughput RL libraries operate using flat, non-hierarchical agents. We next describe the challenges associated with parallelizing hierarchical agents, and our approach to solving them.

## 3. Method

In this section we introduce Scalable Option Learning (SOL), our hierarchical RL method designed to optimize both performance and computational throughput.

### 3.1. Objective

At a high level, we jointly optimize actor-critic objectives (Konda & Tsitsiklis, 1999) for all the options as well as the controller, each of which consists of a policy loss, a value loss, and an exploration loss. In addition to the controller

policy $\pi_\Omega$ and option policies $\pi_\omega$, we learn controller and option value function estimators $\hat{V}_\Omega$ and $\hat{V}_\omega$, which estimate the future returns of the controller and option policies using their respective reward functions. Additional details and exact definitions can be found in Appendix D.1.

**Policy Objective**    The policy objective we optimize is:

$$\mathcal{L}_{\text{policy}} = \mathbb{E}_\tau \left[ -\sum_{t=0}^{\infty} \sum_{\omega \in \Omega} \left( \delta_{z_t = \pi_\Omega} \log \pi_\Omega(\omega | s_t) A^{\text{task}}(s_t, \omega) \right. \right.$$

$$\left. \left. + \delta_{z_t = \pi_\omega} \log \pi_\omega(a_t | s_t) A^\omega(s_t, a_t) \right) \right]$$

where $\tau = (s_0, z_0, a_0, r_0, s_1, z_1, a_1, r_1, ...)$ represents trajectories generated by the agent. The variable $z_t$ represents the policy being executed at time $t$, which can be either the controller $\pi_\Omega$ or any of the option policies $\pi_\omega$, and $\delta$ represents a one-hot indicator. Here $A^{\text{task}}$ represents the advantage associated with the controller $\pi_\Omega$ and task reward $R$, and $A^\omega$ represents the advantage associated with the option policy $\pi_\omega$ and option reward $R_\omega$. On timesteps where the controller selects a new option, no environment-level action is taken, so a duplicated state is inserted to be acted upon by the newly selected option on the next timestep.

**Value Objective**    Let $\hat{V}_\Omega$ and $\hat{V}_\omega$ denote the agent's parameterized value estimates of the task reward $R$ and option reward $R_\omega$ respectively. Our value objective is:

$$\mathcal{L}_{\text{value}} = \mathbb{E}_\tau \left[ \sum_{t=0}^{\infty} \left( \delta_{z_t = \pi_\Omega} (V_\Omega(s_t) - \hat{V}_\Omega(s_t))^2 \right. \right.$$

$$\left. \left. + \sum_{\omega \in \Omega} \delta_{z_t = \pi_\omega} (V_\omega(s_t) - \hat{V}_\omega(s_t))^2 \right) \right]$$

The definitions of the option advantage and value functions $A^\omega$ and $V^\omega$ do not depend on any of the other options or the controller, however their estimators are trained on the distribution of states induced by the entire system. This is designed to train options using their respective rewards independently of one another, on a state distribution that is wider than the start state distribution. At first glance, it might be unclear how to estimate $V^\omega$ from trajectories where different option behaviors are mixed, since it depends on an infinite sum of rewards induced by following a single option $\pi_\omega$. We address this by applying the following recurrence relation and approximation:

$$V^\omega(s_t) = \mathbb{E}_{\pi_\omega} [\sum_{k=0}^{\infty} \gamma^k R_\omega(s_{t+k}, a_{t+k}) | s_t]$$

$$\approx \mathbb{E}_{\pi_\omega} [\sum_{k=0}^{K} \gamma^k R_\omega(s_{t+k}, a_{t+k}) + \gamma^{K+1} \hat{V}^\omega(s_{t+K+1}) | s_t]$$

Here $K$ is the remaining number of steps the current option $\omega$ is executed for before a different option is called. During training, we approximate this expectation with a $K$-step return, and the resulting scalar is then used as a target for $\hat{V}^\omega(s_t)$. Note that this differs from the standard bootstrapping schemes used in actor-critic methods like PPO, which always use a fixed rollout length for value estimation.

**Exploration Objective**    We apply standard entropy bonuses on both the controller and option policies to encourage local exploration (Williams & Peng, 1991; Mnih et al., 2016; Schulman et al., 2017).

**Adaptive Option Length**    Each time the controller is called, in addition to the next option $\omega$ to execute it also outputs the number of time steps $l \in L = \{2^i : 1 \leq i \leq M\}$ to execute $\omega$ for, via a softmax over $L$. This can be viewed as operating with an augmented option set $\bar{\Omega} = \Omega \times L$, where the reward for the controller action $(\omega, l)$ at $s_t$ is $\sum_{k=t}^{t+l} R(s_k, \pi_\omega(s_k))$. The length selection head is trained along with the option selection head using the controller reward. Note that for a given $\omega$, choosing higher $l$ leads to higher immediate controller reward since more timesteps are being summed over, but less flexibility in switching to a different option $\omega'$ based on new observations. By setting option lengths to be exponentially spaced, we are able to cover a wide range of values. In Appendix E.2, we show that the controller learns to effectively tune $l$, and we study the effect of varying the number of possible option lengths $M$ in Appendix E.4.

Our global objective is the sum of the policy, value, and exploration objectives, and is trained online using the trajectories generated by the agent.

### 3.2. Discussion

We next discuss some properties of our objective and how it addresses certain known challenges in hierarchical RL. The first is option degeneracy, a phenomenon observed in prior works (Harb et al., 2018; Luo et al., 2023) where the controller always chooses options to have length one. We did not observe this undesirable behavior empirically (see Appendix E.2) and next give justification why our objective avoids it.

Suppose there is a state $s$ at which it is optimal for the controller to execute option $\omega$ for $k$ steps. Now note that calling $\omega$ either once for $k$ steps, twice for $k/2$ steps, four times for $k/4$ steps, and so on will all give the same expected reward to the controller (up to discount factors). Therefore, if the controller has experienced sufficient trajectories with each of these sequences, its entropy bonus will encourage it to choose $(\omega, k), (\omega, k/2), (\omega, k/4), ...$ at state $s$ with equal probability, and not collapse to the shorter option lengths.

A second known challenge in hierarchical RL is that of non-stationarity (Klissarov et al., 2025a), which can arise when different components of the hierarchical agent depend on each other and thus create targets that change over the course of training. An appealing feature of our objective is that it limits the dependence of each option policy on both the controller policy as well as the other option policies, which reduces non-stationarity. More precisely, for some state $s$, action $a$ and option $\omega$, the advantage $A^\omega(s, a)$ and value $V^\omega(s)$ do not depend on what $\pi_\Omega$ or any other $\pi_{\omega'}$ do after $\pi_\omega$ is finished executing. The only influence they have on $\pi_\omega$ is through the starting state distribution they induce, i.e. how often $s$ is visited in the first place—this is a desirable feature since it means $\pi_\omega$ will be trained on a broader state distribution than the initial state distribution of the MDP, which might be narrow.

### 3.3. Scaling Challenges

Before discussing the details of our system design, it is important to understand why scaling hierarchical RL methods is not straightforward. Hierarchical systems execute a sequence of policies, chosen from $\Pi_\Omega = \{\pi_\Omega\} \cup \{\pi_\omega\}_{\omega \in \Omega}$, which depends on the observations. Because of this, in a batch of trajectory segments of size $B \times T$, any given slice of size $B$ at time $t$ will likely correspond to several different policies, with correspondingly different reward functions (see Figure 1). As a result, both the forward passes through the policy network, which are needed to compute the action probabilities and value estimates, as well as the return or advantage computations, which operate on the different option and controller rewards, are difficult to parallelize. Current hierarchical methods such as (Nachum et al., 2018; Gürtler et al., 2021; Levy et al., 2017; Klissarov et al., 2017) process a single trajectory at a time, which is sufficient for the continuous control settings where they are tested that require a relatively small number of samples (in the millions). However, complex, open-ended environments such as NetHack typically require billions of samples, which in turn requires more scalable hierarchical methods.

### 3.4. System Design

We address these challenges through three design choices: i) a single neural network with multiple action heads and an in-

dexing vector which represents both high and low-level policies, ii) an environment wrapper in the actor workers which tracks active policies and computes corresponding rewards, and iii) efficient parallelized masking when computing the advantages and value targets for each policy. These enable leveraging existing high-throughput RL libraries such as Sample Factory (Petrenko et al., 2020). We provide pseudocode below, a system overview in Figure 1, and describe each component in detail in the remainder of this section.

```
1  # initialize original env
2  env = gym.make(...)
3  # the SOL wrapper augments the env's
4  # observation space with an index
5  # indicating the active policy and handles
6  # its switching during the step() function.
7  # It also augments the action space with
8  # controller actions.
9  env = SOLWrapper(env)
10 envs = make_batched_envs(env)
11
12 # create single network with augmented
13 # input and action space
14 policy_net = create_network(
15     input_space = env.obs_space,
16     action_space = env.action_space,
17     ...
18 )
19
20 # training loop
21 for n in 1:num_updates:
22     # get trajectory batch from
23     # actor workers (can be done async)
24     # obs: observations size BxTxD
25     # rew: mixed rewards size BxT
26     # done: terminals size BxT
27     obs, rew, done = get_traj_batch(
28         policy_net,
29         envs
30     )
31     # controller rewards depend on the
       future and must be computed here
       rew = fill_controller_rewards(rew, obs)
32     # compute advantages and value targets
33     # for all policies simultaneously.
34     adv, val = sol_compute_adv_values(
35         rew, done, obs.policy_indx
36     )
37     # update with any policy gradient
38     # objective (e.g. PPO)
39     policy_net = policy_gradient_update(
40         obs, adv, val, policy_net
41     )
42
```

*Listing 1.* SOL pseudocode. See our code release for full details.

**Architecture** In order to process trajectory batches efficiently in parallel, we adopt a single neural network architecture which represents all the option policies as well as the controller policy. In addition to the environment observation, the network receives a one-hot vector $u$ of dimension $|\Omega|+1$ which indicates which of the policies in $\{\pi_\omega\}_{\omega \in \Omega} \cup \{\pi_\Omega\}$

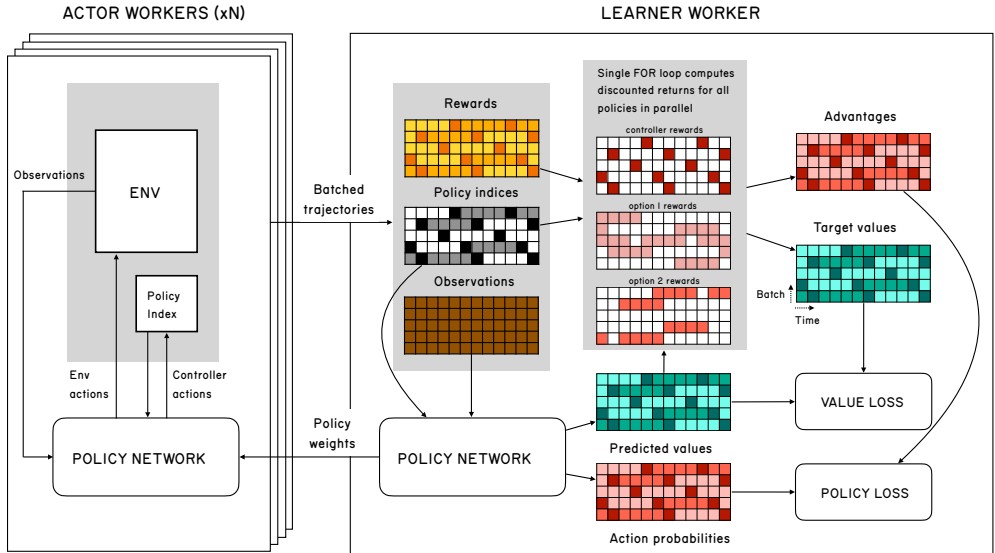

*Figure 1.* System overview. We use a single network with an augmented action space to represent both option policies and the controller policy, enabling batched forward passes for all policies at once. In the actor workers, a modified environment wrapper tracks the policy index based on the high-level controller actions, and routes the low-level actions to the environment. The learner worker continuously updates the policy with batched forward passes through the policy network and efficient tensorized return computations. Different shades indicate quantities associated with different policies: darkest is the controller, the other two are options $\omega_1$ and $\omega_2$. All grids (rewards, policy indicies, advantages, values...) are of size $B \times T$: rows are batch indices, columns are time steps.

to represent. The network's output space is $\mathcal{A} \times \Omega \times L$, where $L$ is the set of possible option lengths. For each input observation, the network outputs three distributions: a distribution over environment actions $\Delta(\mathcal{A})$, a distribution over options $\Delta(\Omega)$, and a distribution over option lengths $\Delta(L)$. If $u$ represents one of the option policies $\pi_\omega$, then $\Delta(\mathcal{A})$ is kept and sent to the environment wrapper which we describe next. Otherwise if $u$ represents $\pi_\Omega$, the distributions over options and option lengths $\Delta(\Omega), \Delta(L)$ are sent instead. These modifications only add a small number of extra parameters (+3%) relative to the flat network architecture. See Appendix D.2 for an illustration and more details.

**Actor Workers**   The second component is an environment wrapper in each actor worker which tracks which policy is currently being executed, switches them based on the termination conditions and controller actions, computes option rewards, and duplicates the last observation whenever the controller policy is called. Specifically, a variable p tracks which policy is currently active, and at each time step is converted to the one-hot vector $u$ which is fed as input to the network in addition to the observation. Each time an option terminates after being executed for the number of steps last output by the controller, p changes based on the controller distribution $\Delta(\Omega)$ output by the network, and the next option length $l \sim \Delta(L)$ is recorded in the environment wrapper. Otherwise, an action is sampled from $\Delta(\mathcal{A})$ and routed to the environment instance. The rewards for each

of the option policies $\pi_\omega$ are computed using $R_\omega$ from the observation directly. The reward used to train the controller policy $\pi_\Omega$, which is the sum of the task rewards for the option that it calls, depends on the future execution of that option and is computed in the learner thread. Pseudocode providing more details is included in Appendix D.3.

**Learner Worker**   Given the above two components, the main learner process which updates the policy network receives the following tensors: observations $O$ of size $B \times T \times D$ (where $D$ is the observation dimension), rewards $R$ of size $B \times T$, episode terminations of size $B \times T$, and one-hot policy indices $P$ of size $B \times T \times |\Omega| + 1$. Note that the rewards in $R$ are of mixed types: $R_{ij}$ is of the type corresponding to the policy $P_{ij}$. The first step is to fill in the rewards $R$ corresponding to the controller policy, which could not be previously computed in the actor threads since they depend on the future execution of the option policy that the controller calls. This is done with a single FOR loop compiled to C using Cython (Behnel et al., 2011). Next, for each policy indexed by $P$, the learner process must compute two main quantities: the empirical returns and the advantages. By using tensorized operations and caching various intermediate quantities for each policy, we are able to compute both of these quantities for all policies simultaneously, using their respective rewards, with a single backward FOR loop over the time dimension $T$. At a high level, this is done by: i) tracking which policies have had their bootstrapped

| Method | Hierarchical | Steps/second |
|---|---|---|
| HIRO | Yes | 55 |
| Option-Critic | Yes | 75 |
| Hierarchical RLLib | Yes | 780 |
| Multiple Option-Critic | Yes | 1200 |
| SOL | Yes | 43000 |
| APPO | No | 50000 |

*Table 1.* Throughput comparison on the NLE. SOL achieves 35x-580x speedup compared to prior hierarchical methods.

values already added to the cumulative returns, ii) at each time step $t$, adding the rewards $R[:, t]$ to the appropriate policy returns based on the current policy indices $P[:, t]$, and iii) appropriately handling episode terminations for each policy, based on the last observation during which it is executed. Finally, V-trace off-policy corrections are applied to both the controller and option policies, to account for potential lags between the actor and learner workers in asynchronous settings. See Appendix D.4 for source code with full details.

**Throughput Comparison**  We instantiate our algorithm using the high-throughput Sample Factory library (Petrenko et al., 2020). In Table 1 we compare the throughput of our algorithm with that of public implementations of four other hierarchical algorithms: HIRO (Nachum et al., 2018), Option-Critic (Bacon et al., 2017), Multiple Option Critic (MOC) (Klissarov & Precup, 2021) and the hierarchical training implemented in RLLib (Wu et al., 2021). For HIRO and Option-Critic we use the same NLE encoder as in our experiments in Section 5, for MOC and RLLib we used a visual rendering pipeline instead for code compatibility reasons (SOL evaluated with the same pipeline gets similar throughput as with our standard encoder). We used the same hardware for all comparisons and tuned the batch size and number of environments to obtain the best throughput. Additional details can be found in Appendix B.4. Our algorithm is ~35-580× faster than the other four hierarchical methods, and retains 86% of the speed of the flat agent.

We note that the design decisions above are not library-specific and our algorithm could be instantiated with other implementations which use asynchronous actor workers to collect experience and a learner worker to perform batch policy updates on the GPU, which is a common design in distributed RL (Espeholt et al., 2018; Küttler et al., 2019; Mella et al., 2022; Suarez, 2024).

## 4. Related Work

Early hierarchical methods focus on the tabular setting, such as Hierarchical Q-learning (Wiering & Schmidhuber, 1997; Singh, 1992a;b; Kaelbling, 1993) and feudal RL (Dayan & Hinton, 1992).

Options (Sutton et al., 1999) provide a general framework for temporally extended decision-making. The original work considers methods for learning value functions or models over a set of hardcoded options in the tabular setting. Several follow-up works have explored learning options instead, for example by identifying bottleneck states (McGovern & Barto, 2001; Stolle & Precup, 2002; Menache et al., 2002; Şimşek & Barto, 2008) and defining rewards based on reaching them. The Option-Critic architecture (Bacon et al., 2017) jointly learns the option policies with a value function over options using only the task reward. However, the benefits were primarily in transfer to new tasks, and it was not shown to clearly improve over a flat policy on the original task.

Several hierarchical methods based on deep RL have been proposed and evaluated on continuous control environments, such as HIRO (Nachum et al., 2018), HAC (Levy et al., 2017), and HiTS (Gürtler et al., 2021). While improving over flat policies, they focus on sample efficiency through off-policy learning, rather than scaling to large numbers of samples. As a result, implementations are single-process and not designed to scale to billions of samples.

Closer to our work are hierarchical variants of PPO (Schulman et al., 2017), which include HiPPO (Li et al., 2020), PPOC (Klissarov et al., 2017) and MOC (Klissarov & Precup, 2021). Like SOL, these are policy gradient algorithms, but they differ in their objectives: they do not use our value bootstrapping or adaptive option length selection schemes, and attempt to learn using the task reward alone. PPOC and MOC use the relatively optimized OpenAI baselines library (Dhariwal et al., 2017) which uses parallelized experience collection, and are an order of magnitude faster than the above methods. However, MOC is still over an order of magnitude slower than ours.

Feudal Networks (Vezhnevets et al., 2017) are another deep RL architecture, which is conceptually similar to HIRO but defines goals in embedding space rather than the original state space. This work reported promising results, however the official code has not been released and we were unable to find third-party reimplementations which reproduced their results, making comparisons difficult.

Our approach is related to the joint skill learning described in MaestroMotif (Klissarov et al., 2025b), but differs in several ways: i) we learn our controller jointly with the options, whereas they use a frozen LLM, ii) we do not use hardcoded initiation and termination conditions, and iii) we use separate value function bootstrapping across the different options and controller policy.

Similar to SOL, Agent57 (Badia et al., 2020) achieves high throughput by indexing a family of policies with a shared neural network using a one-hot vector, which are then selected by a controller trained to maximize the task reward.

However, each policy is executed for the full episode, unlike SOL which switches between policies throughout.

There are a number of works on hierarchical agents for robotics and embodied AI (Heess et al., 2016; Peng et al., 2017; Yokoyama et al., 2023; Szot et al., 2021; Qi et al., 2025; Pertsch et al., 2020; Chen et al., 2023) which learn each option (or skill) separately, and subsequently train a high-level controller to coordinate them. This approach has been shown to be effective in real and simulated robotics settings. However, learning each skill independently requires a starting state distribution that is sufficiently diverse, which may not be the case when the appropriate states may only be reached by mastering and coordinating other skills (see for example Klissarov et al. (2025b)).

Park et al. (2025) also study hierarchical RL at scale, but in the offline setting, whereas we focus on developing scalable methods for the online setting. Their results also show the limits of naively scaling flat policies, further highlighting the need for scalable hierarchical methods.

# 5. Experiments

We evaluate our proposed approach across three environments: MiniHack, NetHack, and Mujoco. The NetHack Learning Environment (NLE) (Küttler et al., 2020) is based on the notoriously difficult roguelike game of NetHack, which requires the player to descend through many procedurally generated dungeon levels to recover a magical amulet. The game involves hundreds of object and monster types, and succeeding requires mastering many capabilities including exploration, combat, resource management and long-horizon reasoning, with successful episodes often lasting $10^4 - 10^5$ steps (Paglieri et al., 2025). MiniHack (Samvelyan et al., 2021) is a framework based on NetHack which enables easy design of RL environments, allowing the targeted testing of agent capabilities.

In the next two sections, we assume access to a given set of option intrinsic rewards $R_\omega$, which we specify. We also investigate automatically discovering $R_\omega$ in Section 5.3.

We consider the following methods in our comparisons:

- APPO (task reward): a flat asynchronous PPO agent trained with task reward only.

- APPO (task+option rewards): a flat APPO agent trained with a linear combination of task reward and option rewards $R_\omega$, with coefficients optimized by grid search.

- SOL+HiPPO: an instantiation of HiPPO (Li et al., 2020) using our scalable framework. This uses only the task reward and no option rewards.

- SOL: our hierarchical agent.

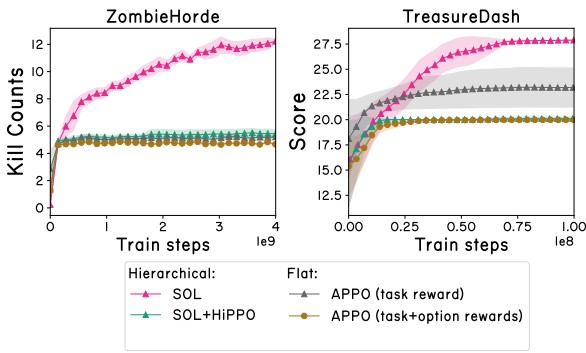

*Figure 2.* Results on MiniHack. Shaded regions represent two standard errors over 10 seeds.

This set of comparisons allows us to disentangle the effect of the hierarchical architecture from benefits due to prior knowledge in the form of option rewards. APPO (task+option rewards) has access to the the same additional option rewards as SOL, and incorporates them with a flat architecture. SOL has access to option rewards, and makes use of them through a hierarchical architecture. SOL+HiPPO has a hierarchical architecture, but does not use option rewards. We did not include the prior hierarchical RL methods shown in Table 1, since they would require an intractably long time to process the same number of samples as SOL. An exception is MOC, which we were able to run on Mujoco. This was possible because MOC is the fastest among prior methods and Mujoco requires much fewer samples than MiniHack and NetHack.

We additionally include Motif (Klissarov et al., 2024), a method which uses an LLM to synthesize intrinsic rewards, in our NetHack experiments since it is the current state of the art, but note that it is orthogonal and can be combined with our method, for example by adding its rewards to one or more of our option rewards. We include this variant in our comparisons under the name SOL+Motif. Motif also makes different assumptions: it requires an LLM and observations with a meaningful textual component, hence it cannot be directly applied to environments like Mujoco. Full experiment details, including architectures and hyperparameters, can be found in Appendix B.2.

## 5.1. MiniHack and NetHack

In addition to NetHack, we design two MiniHack environments which specifically test the ability to perform difficult credit assignment and coordinate different behaviors, while also being fast to run. These environments are described next, with full details in Appendix C.1.

**ZombieHorde** The agent is initialized in a room with a horde of zombies it must defeat, which also contains a safe temple area the zombies cannot enter. The zombies

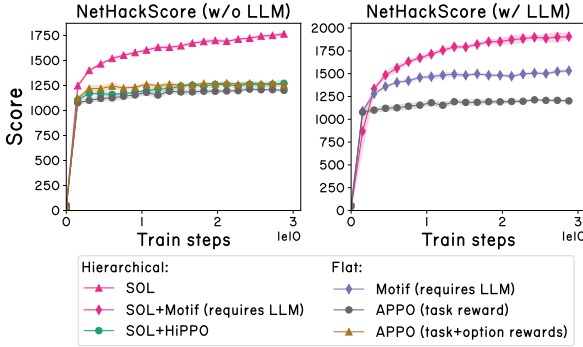

*Figure 3.* Results on NetHack. Shaded regions represent two standard errors over 5 seeds. In the right plot, APPO does not use an LLM and is included as a reference.

are too numerous to fight at once, however, and the agent must periodically retreat to the temple to heal whenever its health becomes too low. This poses a challenging credit assignment problem due to delayed rewards: healing takes dozens of time steps while giving no rewards, but leads to much higher rewards in the long term since it allows the agent to survive future fights. The option rewards here are $R_{\omega_1} = \Delta \texttt{Score}, R_{\omega_2} = \Delta \texttt{Health}$, indicating the per-timestep changes in agent score (which increases for each zombie destroyed) and hit points. For hierarchical methods, the controller's reward is also $\Delta \texttt{Score}$.

**TreasureDash** The agent is initialized in a hallway filled with piles of gold next to a staircase. The agent has a small number of timesteps in which it can choose to gather gold for a small amount of reward or descend the stairs for a large one-time reward and episode termination. The optimal strategy is to gather as much gold as possible in the given time before descending the stairs on the final timestep. Agents that fail to balance the two competing sources of reward can fall into the local optima of either immediately descending the stairs or gathering gold until the timer runs out. The option rewards are $R_{\omega_1} = \texttt{AtStairs}, R_{\omega_2} = \Delta \texttt{Gold}$, and the controller optimizes total reward.

**NetHackScore** We use the NetHackScore environment from the NLE paper, with the modified EAT action used in Klissarov et al. (2024). The game score serves as the task reward function. The option rewards we consider here are $R_{\omega_1} = \Delta \texttt{Score}, R_{\omega_2} = \Delta \texttt{Health}$. Hierarchical methods also use $\Delta \texttt{Score}$ as their controller reward.

**Results** On ZombieHorde (Fig. 2, left), all baselines quickly saturate after ~5 kills, while SOL keeps improving and achieves a significantly higher final performance. On TreasureDash (Fig. 2, right), SOL achieves close to the optimal performance of 28 points, whereas the other methods are suboptimal to various degrees (obtaining 20

points for only gathering gold or going to the stairs, 23 for gathering 3 gold before exiting through the stairs). In both cases, continued training of the baselines does not result in higher performance, illustrating that scaling alone can be insufficient for tasks involving hard credit assignment. On NetHackScore, we train all agents for 30 billion steps. In Figure 3 (left) we compare LLM-free methods: both flat APPO agents perform similarly, indicating that adding intrinsic rewards (Health) to the task reward (Score) is not helpful here. This can be explained by the fact that rewarding the change in health discourages the agent from engaging in combat (which causes loss of health) and exploring. SOL+HiPPO converges to similar performance as the two flat agents, indicating it is not able to leverage hierarchical structure. SOL steadily improves and achieves higher performance than the other agents. In Figure 3 (right) we compare methods which leverage LLMs. Motif improves over the flat APPO baseline, consistent with prior work. Our method SOL+Motif, which adds Motif rewards to its Score option, significantly improves on Motif and sets a new state of the art on NetHackScore. In Appendix E.1, we repeat these experiments with two other NetHack characters different from the default Monk, and find that these trends are maintained. It is notable that SOL's performance still appears to be increasing after 30 billion steps (~2 weeks of training), suggesting the that benefits of scale unlocked by our method remain to be fully realized.

In Appendix E.2, we include visualizations which shed light on SOL's behavior. In particular, we find that: i) options are able to optimize their respective rewards and yield qualitatively different yet complementary behaviors, ii) the controller is able to coordinate the options and call them in a state-dependent manner, and iii) the controller is able to adapt the option execution length based on the task and option. We also include ablations of our value function bootstrapping scheme (E.3), our adaptive option length mechanism (E.4), option reward scaling (E.5), the effect of adding redundant or unhelpful options (E.6), and improving sample efficiency through off-policy updates (E.7).

### 5.2. Continuous Control

To test our algorithm's generality, we next consider continuous control mazes provided by Gymnasium Robotics (de Lazcano et al., 2024). We first found that flat APPO agents were able to solve all existing PointMaze environments (U-Maze, Medium and Large) when trained sufficiently, indicating that these mazes do not require hierarchy in the large sample regime (details in Appendix E.8). We therefore designed a more challenging maze called the G-maze, described in Appendix C.3. The agent is initially close to the goal in a G-shaped maze but separated by a wall, and the optimal trajectory requires an initial increase in distance followed by a larger decrease. This creates a

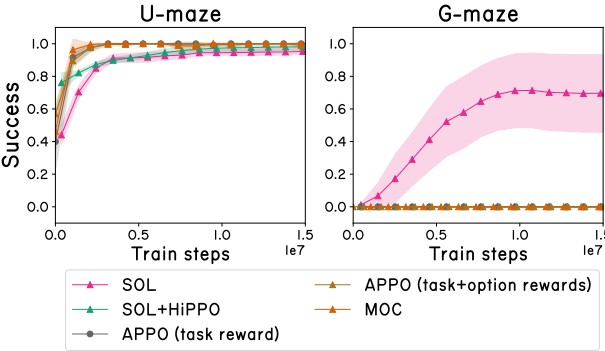

*Figure 4.* Results on two PointMaze layouts. Shading represents two standard errors over 10 seeds.

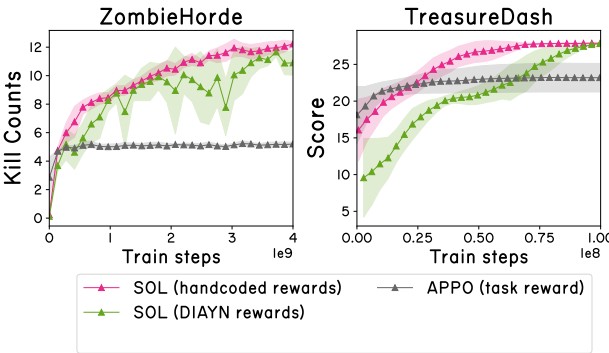

*Figure 5.* Results on MiniHack. Shaded regions represent two standard errors over 5 seeds.

local optimum in the reward landscape that is difficult to escape. We choose option rewards $R_\omega$ to be the velocity in the positive and negative $x$ and $y$ directions, which are already provided as part of the state, as well as the task reward, for a total of 5 options. Results for both the U-Maze and our G-maze are shown in Figure 4. On the U-Maze, all agents are able to achieve high success rates. However, on the G-maze, SOL is the only method able to make progress, achieving roughly 70% success while the others remain at zero. This provides evidence for SOL's generality.

### 5.3. Automatically Learning Option Rewards

We next provide an experiment illustrating that SOL is compatible with methods for automatic reward synthesis and can be used to experiment with them at scale. We define a variant SOL+DIAYN where the option rewards $\{R_\omega\}_{\omega \in \Omega}$ are learned using DIAYN (Eysenbach et al., 2019), a method for automatic skill discovery. DIAYN operates by training a discriminator online to classify different policies based on the state, while the policies are trained using the discriminator's class probabilities as rewards. This has the effect of producing policies which visit distinct states, enabling them to be distinguished by the discriminator.

In our setup, a discriminator $D : \mathcal{S} \to \Delta(\Omega)$ is trained online to classify the $|\Omega|$ different option policies. One option reward is set to be the task reward, and each remaining option $\omega$ has reward given by $R_\omega(s) = p_D(\omega|s)$. As before, the controller's reward is the task reward. The discriminator is trained and assigns option rewards fully on the GPU in the learner thread, causing a drop in throughput of only ~7%. Full experimental details are in Appendix B.3. Results on both MiniHack environments are shown in Figure 5. In both cases, SOL+DIAYN learns more slowly than SOL but converges to similar final performance, significantly outperforming the flat baseline while requiring no prior knowledge in the form of option rewards.

### 5.4. Discussion

Our experimental results point to several takeaways. First, for the difficult credit assignment tasks we consider, flat agents struggle to escape suboptimal local minima, even when equipped with prior knowledge in the form of intrinsic rewards. Second, hierarchical structure alone is not sufficient either, as illustrated by the fact that SOL+HiPPO, which is trained with the task reward only, does not outperform flat agents. This is consistent with other works which have reported that hierarchical agents trained with the task reward alone have difficulty outperforming flat baselines (Bacon et al., 2017; Smith et al., 2018). Our best performing agent SOL combines both hierarchical structure with useful option rewards, which can be derived from prior knowledge or learned, that reflect the optimal policy in certain parts of the state space. This suggests that for certain classes of hard credit assignment problems, both hierarchy and good intrinsic rewards are necessary to unlock each others' benefits.

## 6. Conclusion

This work introduces, to our knowledge, the first online hierarchical RL algorithm which is able to scale to billions of samples. Its performance and scalability are enabled by algorithmic and systems-level features which enable efficient GPU parallelization. When trained at scale on the challenging NLE, SOL surpasses flat baselines and learns options with different behaviors which are effectively coordinated by the controller. It also proves effective in continuous control and MiniHack environments, showcasing its generality. We release our code to facilitate future research on long-horizon decision-making at scale.

## Impact Statement

This paper presents work whose goal is to advance the field of Reinforcement Learning. There are many potential societal consequences of our work, none which we feel must be specifically highlighted here.

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

# A. Limitations and Future Work

First, while we provide evidence that SOL is compatible with both hardcoded and learned option rewards, the question of how best to learn option rewards remains an open area of research, and more work is needed to find truly general methods which are successful across a wide range of environments. There are several potential ways to generate option rewards automatically, which constitute interesting future work. These include: using diversity measures (Eysenbach et al., 2019) to define rewards which induce a diverse set of option policies (we we explore in this work), novelty bonuses (Burda et al., 2019; Henaff et al., 2022) which could encourage exploratory options, distances to goals output by the controller (Nachum et al., 2018; Vezhnevets et al., 2017), or using LLMs synthesize rewards via code generation or preference ranking (Klissarov et al., 2024; 2025b; Ma et al., 2023; Kwon et al., 2023b;a; Fan et al., 2022).

Second, our system is designed for computational efficiency, not sample efficiency. It focuses on achieving superior asymptotic performance in the large sample regime, rather than making optimal use of limited samples. Therefore, it is currently limited to settings where samples are easy to gather and compute is the bottleneck, such as video games, digital agents or sim-to-real transfer. Some of the design decisions, such as using a single neural network to represent both high and low-level policies, and the parallelized return computations, could in principle be incorporated into a model-based RL framework, which could potentially improve sample efficiency.

# B. Experiment Details

## B.1. Architectures

For all MiniHack and NetHack experiments, we used a neural network architecture which mostly follows the Chaotic Dwarven GPT5 architecture of (Kanervisto & Jucys, 2022) with one change: we replaced the pipeline which renders glyphs to pixel images and runs them through an image-based convnet with a direct glyph embedding layer followed by 2 convolutional layers. We found this reduced the memory footprint (allowing us to have a larger PPO batch size) while giving slightly better performance. The pipelines processing the messages and bottom-line statistics (blstats) were unchanged. Specifically, the blstats are processed by a two-layer MLP with 128 hidden units at each layer, and the message character values are divided by 255 and also processed by a 2-layer MLP. The embeddings for the glyph images, blstats and messages are then concatenated and passed to a recurrent GRU (Cho et al., 2014). For the hierarchical models, we embed the policy index to a 128-dimensional vector which is concatenated with the other embeddings before passing to the GRU. This same vector is also replicated and added to all spatial locations in the glyph image crop. We also include an extra linear layer mapping the last hidden layer to controller actions.

For Mujoco experiments, we used a 2-layer network with 64 hidden units at each layer and tanh activations. The network outputs the mean and variance of a Gaussian distribution over actions, whose dimension is that of the action space. For hierarchical agents, the policy one-hot is concatenated with the input. As before, we add an extra linear layer mapping the last hidden layer to controller actions. The observation includes the agent's $(x, y)$ position as well as the desired goal position. We do not use a GRU for Mujoco experiments since the environment is fully observed.

## B.2. Hyperparameters

For all NLE agents, we used the common PPO hyperparameters which are listed in Table 2. Our SOL agents additionally use the hyperparameters listed in Table 5.

*Table 2.* Common PPO Hyperparameters for different environments. The same set of hyperparameters are used for MiniHack and NetHack, a different set is used for Mujoco PointMaze.

| Hyperparameter | MiniHack&NetHack | Mujoco PointMaze |
|---|---|---|
| Rollout length | 1024 | 256 |
| GRU recurrence | 256 | none |
| GRU layers | 1 | none |
| PPO epochs | 1 | 10 |
| PPO clip ratio | 0.1 | 0.2 |
| PPO clip value | 1.0 | 1.0 |
| Encoder crop dimension | 12 | N/A |
| Encoder embedding dimension | 128 | N/A |
| Reward Scaling | 0.01 | 10 |
| Exploration loss coefficient | 0.003 | 0.001 |
| Exploration loss | entropy | entropy |
| Value loss coefficient | 0.5 | 0.5 |
| Max gradient norm | 4.0 | 0.1 |
| Learning Rate | 0.0001 | 0.003 |
| Batch size | 32768 | 32768 |
| Worker number of splits for double-buffering | 2 | 2 |
| V-trace $\rho$ | 1.0 | 1.0 |
| V-trace $c$ | 1.0 | 1.0 |
| Discount factor $\gamma$ | 0.99 | 0.99 |

*Table 3.* `APPO (task/task+option rewards)` Hyperparameters

| Environment | Hyperparameter | Value | Values swept |
|---|---|---|---|
| MiniHack-ZombieHorde | Score reward scale | 1 | 1 |
| | Health reward scale | 1 | 0, 1, 3, 10, 20 |
| MiniHack-TreasureDash | Stairs option reward scaling | 1 | 1 |
| | Gold option reward scaling | 0.1 | 0, 0.1, 0.3, 1, 3, 10 |
| NetHackScore | Score option reward scaling | 1 | 1 |
| | Health option reward scaling | 1 | 0, 1, 3, 10 |
| PointMaze-GMaze | True goal reward scaling | 1 | 1 |
| | Goal option reward scaling | 1 | 0, 0.01, 0.1, 1, 10 |

*Table 4.* SOL Hyperparameters. The controller extra exploration loss scaling is a factor which is used to further scale the exploration loss coefficient from Table 2 applied to the controller outputs. We found that having this greater than 1 was sometimes helpful.

| Environment | Hyperparameter | Value | Values swept |
|---|---|---|---|
| MiniHack-ZombieHorde | Controller extra exploration loss scaling | 1 | 1, 3, 10 |
| | Controller reward scaling | 0.001 | 0.001, 0.01, 0.1 |
| | Score option reward scaling | 1 | - |
| | Health option reward scaling | 20 | 10, 20 |
| MiniHack-TreasureDash | Controller extra exploration loss scaling | 1 | 1, 3, 10 |
| | Controller reward scaling | 0.001 | 0.001, 0.01, 0.1 |
| | Stairs option reward scaling | 1 | - |
| | Gold option reward scaling | 1 | - |
| NetHackScore | Controller extra exploration loss scaling | 10 | 1, 3, 10 |
| | Controller reward scaling | 0.001 | 0.001, 0.01, 0.1 |
| | Score option reward scaling | 1 | - |
| | Health option reward scaling | 10 | 10, 20 |
| PointMaze-GMaze | Controller extra exploration loss scaling | 1 | 1, 3, 10, 30 |
| | Controller reward scaling | 1 | 0.01, 0.1, 1, 10 |
| | Goal option reward scaling | 1 | - |

*Table 5.* SOL+HiPPO Hyperparameters. We set the number of options to be the same as SOL and swept hyperparameters in the same ranges.

| Environment | Hyperparameter | Value | Values swept |
|---|---|---|---|
| MiniHack-ZombieHorde | Controller extra exploration loss scaling | 1 | 1, 3, 10, 30 |
| | Controller reward scaling | 0.001 | 0.001, 0.01, 0.1 |
| | Number of options | 2 | - |
| MiniHack-TreasureDash | Controller extra exploration loss scaling | 1 | 1, 3, 10, 30 |
| | Controller reward scaling | 0.001 | 0.001, 0.01, 0.1 |
| | Number of options | 2 | - |
| NetHackScore | Controller extra exploration loss scaling | 10 | 1, 3, 10, 30 |
| | Controller reward scaling | 0.001 | 0.001, 0.01, 0.1 |
| | Number of options | 2 | - |
| PointMaze-GMaze | Controller extra exploration loss scaling | 1 | 1, 3, 10, 30 |
| | Controller reward scaling | 1 | 0.01, 0.1, 1, 10 |
| | Number of options | 5 | - |

*Table 6.* Motif Hyperparameters. We trained the reward model using the official source code, data and default hyperparameters. We then trained APPO agents with the same hyperparameters as other agents (Table 2) and tuned the coefficient of the reward model.

| Hyperparameter | Value | Values swept |
|---|---|---|
| LLM reward coefficient | 0.1 (default) | 0.1, 0.3, 1 |

*Table 7.* MOC Hyperparameters. Despite our hyperparameter sweep, results did not change much: MOC worked well on PointMaze-UMaze, and failed to learn on PointMaze-GMaze. Therefore we report results with default hyperparameters.

| Hyperparameter | Value | Values swept |
|---|---|---|
| Number of options | 2 (default) | 2, 4, 8 |
| Learning rate | 0.0008 (default) | 0.001, 0.0003, 0.0001, 0.00008 |
| Probability of updating all options $\eta$ | 0.9 (default) | 0.1, 0.9 |

### B.3. DIAYN Details

Our SOL+DIAYN method includes an additional discriminator network which has an identical trunk as the policy network, followed by a 2-layer MLP with 256 hidden units which outputs a softmax over options $\Delta(\Omega)$. The discriminator uses the same optimizer type and learning rate as the policy network. The DIAYN reward for each option $\omega$ is the discriminator's softmax value corresponding to that option, scaled by a parameter which we list in Table 8. The DIAYN discriminator is updated using the negative log-likelihood loss each time the policy is updated, on the same batch of data.

*Table 8.* SOL+DIAYN Hyperparameters. The discriminator shares the same architecture as the observation encoder, with a 2-layer MLP added with $|\Omega|$ outputs.

| Environment | Hyperparameter | Value | Values swept |
|---|---|---|---|
| MiniHack-ZombieHorde | Controller extra exploration loss scaling | 10 | 1, 3, 10 |
| | Controller reward scaling | 0.001 | 0.001, 0.01, 0.1 |
| | DIAYN Reward Scaling | 0.03 | 0.01, 0.03, 0.1, 0.3, 1.0 |
| | Discriminator Learning Rate | 0.0001 | 0.0001 |
| | Number of options | 3 | 2, 3, 4 |
| MiniHack-TreasureDash | Controller extra exploration loss scaling | 1 | 1, 3, 10 |
| | Controller reward scaling | 0.001 | 0.001, 0.01, 0.1 |
| | DIAYN Reward Scaling | 0.3 | 0.01, 0.03, 0.1, 0.3, 1.0 |
| | Discriminator Learning Rate | 0.0001 | 0.0001 |
| | Number of options | 3 | 2, 3, 4 |

### B.4. Throughput Comparison Details

All experiments were conducted on an NVIDIA V100-SXM2-32GB GPU. We used the same NLE encoder described in Appendix B.1 for HIRO and Option-Critic. We used the following implementations:

- HIRO: https://github.com/watakandai/hiro_pytorch

- Option-Critic: https://github.com/lweitkamp/option-critic-pytorch

- MOC: https://github.com/mklissa/MOC

- Hierarchical RLLib: https://docs.ray.io/en/latest/rllib/hierarchical-envs.html

Other than changing the architecture to process NLE observations, we kept the rest of the hyperparameters at their default values except for the following. We experimented with different batch sizes of off-policy updates for HIRO and Option-Critic, but this did not significantly change the throughput.

For MOC, we increased the number of parallel environments until the throughput saturated, which was 256 here. We used the NLE visual rendering pipeline from (Kanervisto & Jucys, 2022), where NLE glyphs are rendered to pixels and then processed by a standard Atari DQN convolutional encoder. The reason we did this was because the MOC codebase (based on OpenAI Baselines) only supported pixel and continuous vector inputs. We also ran SOL with the same visual rendering pipeline and found that its speed was around $10\%$ faster than the symbolic encoder for the same batch size, hence we do not believe that using a visual rendering pipeline penalizes methods in these comparisons.

### B.5. Compute Details

All experiments were run on single NVIDIA V100-SXM2-32GB GPU machines. For MiniHack experiments, we used 16 CPUs per experiment. Running a job took around 5 hours for TreasureDash and 10 hours for ZombieHorde. For NetHack experiments, we used 48 CPUs per experiment. Running a job for 30 billion steps took around 14 days. For Mujoco, we used 8 CPUs per experiment. Each job ran for less than one day.

### B.6. Other code links

We use the public codebase for our Motif reward model: https://github.com/mklissa/maestromotif. Our codebase is built upon Sample Factory: https://github.com/alex-petrenko/sample-factory, which is licensed under an MIT license.

## C. Environment Details

### C.1. MiniHack

Here we describe the details of our MiniHack environments. Both have a simple action space consisting of 4 movement actions (north, south, east, west) and the `EAT` action. We note that movement also serves to attack: attempting to move on a square occupied by an enemy attacks it.

In ZombieHorde (Figure 6a), the agent 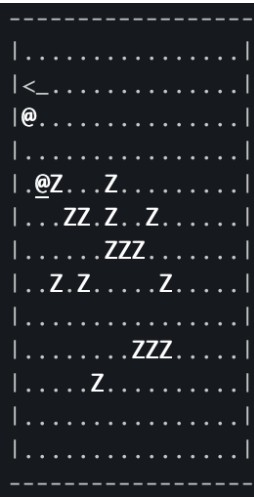 must defeat all the zombies `Z` . Since they are too numerous to fight at once, the agent must periodically retreat to the altar ▬ which the zombies cannot get close to, in order to heal. The priest `@` has no effect here. The time limit is 1500 steps, which enables long periods of healing. Agents in NetHack heal at a rate of about 1 hit point per 10 timesteps, hence full healing can require over 100 steps. Each zombie destroyed gives 20 score points.

In TreasureDash, the agent gets 20 points for exiting through the stairs `>` , which ends the episode. Each piece of gold `$` gives 1 point. The episode time limit is 40 steps. If the agent goes right the whole time, it gathers 20 gold pieces for 20 total points. If it goes left only, it exits and also gets 20 points. The optimal strategy is to gather 8 gold pieces on the right, and then go left all the way to the staircase. This requires stopping the gold-gathering behavior and switching to seeking the staircase.

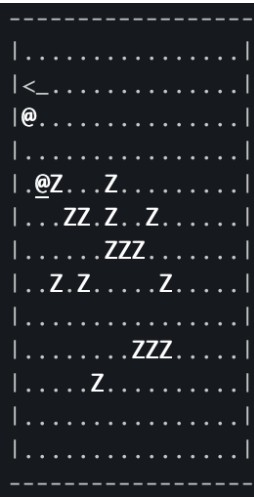

*(a)* ZombieHorde.

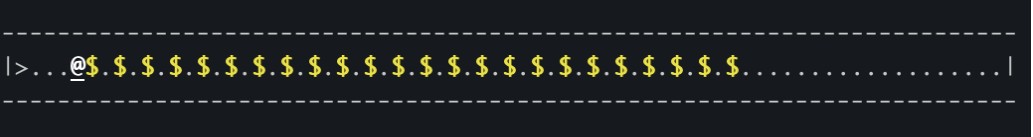

*(b)* TreasureDash

*Figure 6.* MiniHack environments designed to present challenging credit assignment problems.

### C.2. NetHack

The `NetHackScore` environment from the NLE paper includes the following actions: all movement actions, as well as `SEARCH` (needed for finding secret doors, which is often necessary to explore the full level and go to the next), `KICK` (needed for kicking down locked doors, also needed to explore the visit the full level) and `EAT` (needed for eating the comestibles the agent starts with). If the agent does not eat, it will starve before too long which limits the episode length and the maximum progress the agent can make. However, the `EAT` action alone is not enough to eat the comestibles in the agent's inventory, due to the NLE's context-dependent action space. After selecting the `EAT` action, the agent must also select which item in inventory to eat, which requires pressing a key corresponding to the item's inventory slot, which must be included in the original action space, which is often not the case. Therefore, we adopt the modification introduced in (Klissarov et al., 2024), where every time the `EAT` action is selected, the next action is chosen at random from the available

inventory slots given in the message. This is also discussed in Appendix G of their paper.

### C.3. PointMaze

The PointMaze G-Maze environment (shown in Figure 7) uses Gymnasium Robotics and we simply pass the maze map as argument. The agent and goal location are fixed rather than resetting each episode. The reward is the change in euclidean distance between the agent and the goal, and the episode ends whenever the goal is reached according to the default PointMaze criterion. Note that solving this requires the agent to initially move away from the goal, which leads to a decrease in reward, leading to a local optimum that is difficult to escape.

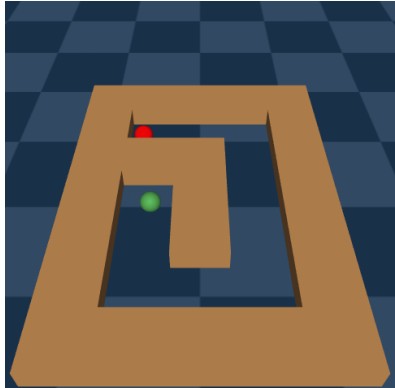

*Figure 7.* PointMaze G-Maze environment.

# D. Algorithm Details

## D.1. Objectives

Here we give the exact definitions of some of the functions used in Section 3.1. Let $\mu : \mathcal{S} \to \Delta(\mathcal{A})$ denote the flattened hierarchical policy, i.e. the mapping from states to actions obtained by executing the options and controller using the call-and-return process. We define the state-option value, state value, and option-advantage functions of $\mu$ associated with the task reward as:

$$Q^{\mu}_{\text{task}}(s_t, \omega) = \mathbb{E}_{\mu}[\sum_{k=0}^{\infty} \gamma^k R(s_{t+k}, a_{t+k})|s_t = s, \omega_t = \omega]$$

$$V^{\mu}_{\text{task}}(s) = \mathbb{E}_{\mu}[\sum_{k=0}^{\infty} \gamma^k R(s_{t+k}, a_{t+k})|s_t = s]$$

$$A^{\text{task}}(s_t, \omega) = Q^{\mu}_{\text{task}}(s_t, \omega) - V^{\mu}_{\text{task}}(s_t)$$

The state-action value, state value, and advantage functions for an option $\omega$ are given by:

$$Q^{\omega}(s_t, a_t) = \mathbb{E}_{\pi_{\omega}}[\sum_{k=0}^{\infty} \gamma^k R_{\omega}(s_{t+k}, a_{t+k})|s_t = s, a_t = a]$$

$$V^{\omega}(s_t) = \mathbb{E}_{\pi_{\omega}}[\sum_{k=0}^{\infty} \gamma^k R_{\omega}(s_{t+k}, a_{t+k})|s_t = s]$$

$$A^{\omega}(s_t, a_t) = Q^{\omega}(s_t, a_t) - V^{\omega}(s_t)$$

In Section 3.1 we noted that calling the controller does not cause the MDP to transition, which means that states in $\tau$ are duplicated each controller call. To illustrate this, let us consider the first time step, with $s_0$ being the first state. First the controller must be called, since we don't know what low-level option to execute. We therefore run $s_0$ through the controller and obtain $\omega_3, l \sim \pi_{\Omega}(\cdot|s_0)$. This means we will execute option policy $\pi_{\omega_3}$ for $l$ timesteps. The first state we must apply it to is still $s_0$, since we haven't passed any actions to the MDP yet. We therefore compute $a_0 \sim \pi_{\omega_3}(\cdot|s_0)$, sample $s_1 \sim p(\cdot|s_0, a_0)$, and repeat this process for $l$ timesteps. We then call the controller again at $s_l$, which produces (for example) $\omega_2, l' \sim \pi_{\Omega}(\cdot|s_l)$, meaning we will execute $\pi_{\omega_2}$ for $l'$ timesteps. Again, since we have not executed any actions in the environment, we then run $s_l$ through $\pi_{\omega_2}$ and the process continues. See Table 9 for an example trace.

## D.2. Neural Network Architecture

SOL's single neural network architecture is shown in Figure 8. A one-hot vector $u$ of dimension $|\Omega| + 1$ indicates which of the policies (among the option policies and the controller policy) to represent. In the actor workers, if $u$ marks the controller, the softmaxes over options and option execution lengths are sampled from and the results are used to update the environment wrapper shown in Appendix D.3. Otherwise, if $u$ marks one of the options, the softmax over environment actions $|\mathcal{A}|$ is sampled from and the sampled action is executed in the environment. In the learner worker, the softmaxes over options and option lengths constitute the action probabilities of the controller and are used to compute the advantage and policy loss at each step it is called. The softmax over environment actions gives the action probabilities of the option policies and is used to compute the advantage and policy loss at any steps that they are called. The scalar value output represents the value estimate of the policy currently marked by the policy indicator $u$, and is used in the learner worker to compute the value loss and advantages of both the controller and option policies. A key advantage of our design is that trajectories can be processed in batch, regardless of which policies are being executed, since they are only differentiated by the policy indicators $u$ which are also fed in as a batch.

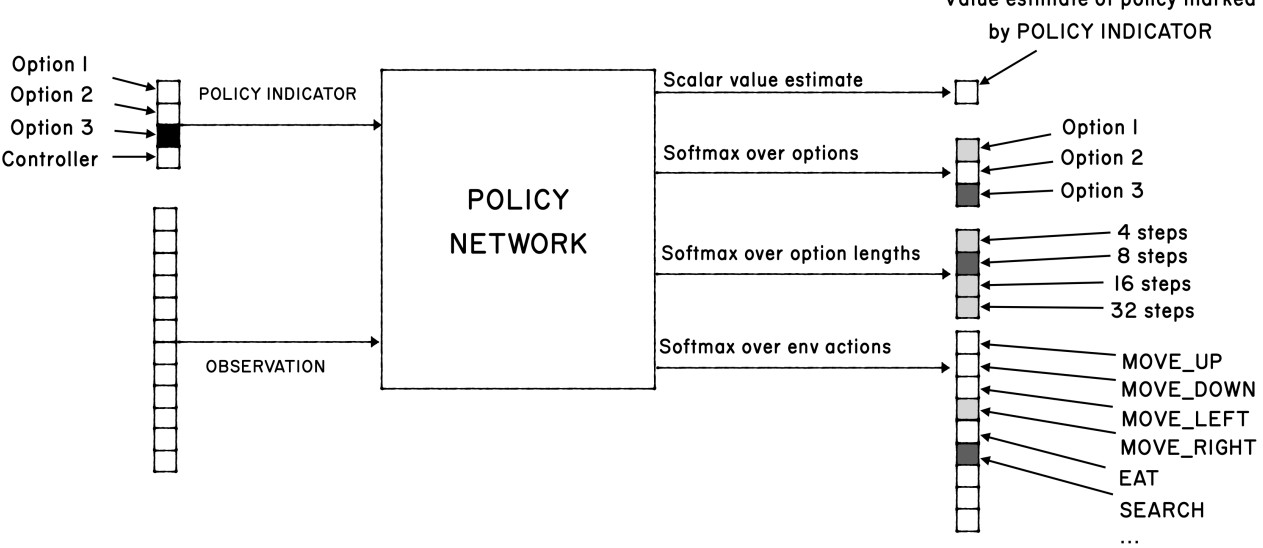

*Figure 8.* SOL's single neural network architecture. The environment actions (bottom right) represent those from NetHack/MiniHack here, but can also be continuous (e.g. in MuJoCo experiments). The set of option lengths in our experiments is $\{1, 2, 4, 8, 16, 32, 64, 128\}$.

SOL only adds a small number of extra parameters on top of the default flat neural network architecture: an extra linear layer mapping the last hidden layer to the controller actions, and an extra embedding layer mapping the policy indicators to a common hidden layer. In our NetHack experiments, this resulted in a network with 4712981 parameters, compared to 4576523 parameters for the default network used by flat agents. This represents only a 3% increase, hence the improvements that we report over the flat agents are unlikely to come from an increased parameter count.

### D.3. Environment wrapper pseudocode

Pseudocode for the environment wrapper in the actor workers is shown below.

```python
class HierarchicalWrapper(gym.Wrapper):

    def __init__(self, env, ...):
        self.env = env
        # update the observation space to include the option
        # and controller rewards as well as policy indices
        self.observation_space = self.env.observation_space + ...
        # update the action space to include actions
        # for option and option execution length selection
        self.action_space = self.env.action_space + ...

        self.option_policies = [...]
        self.option_length = ...

    def compute_option_reward(self, option):
        ...

    def reset(self):
        obs = self.env.reset()
        self.last_obs = obs
        self.current_policy = "controller"
        obs["current_policy"] = self.current_policy
        return obs

    def step(self, action):

        env_action, option_indx, option_length = action

        if self.current_policy == "controller":
            self.current_policy = self.option_policies[option_indx]
            obs = self.last_obs
            obs["current_policy"] = self.current_policy
            self.option_length = option_length
            done = False
            # the controller reward depends on the future, so we compute
            # it in the learner thread and flag for now.
            reward = 42
            self.option_steps = 0
            info = {}
            return obs, reward, done, info
        else:
            obs, done, task_reward, info = self.env.step(env_action)
            self.last_obs = obs
            reward = self.compute_option_reward(obs, self.current_policy)
            self.option_steps += 1

            if self.option_steps == self.option_length:
                self.current_policy = "controller"
            obs["current_policy"] = self.current_policy

            return obs, done, reward, info
```

The wrapper produces trajectories of the form shown below in Table 9. Note that observations are duplicated each time the option changes: they are used first as input to the controller, which chooses the option to execute next, and then the same observation is used as input to the chosen option. The Action row of Table 9 contains both actions of the controller policy, which are options $\omega \in \Omega$, and low-level environment actions of the option policies, which are in the MDP's action space $\mathcal{A}$. The State, Action, Policy Index and Policy Rewards correspond to the $s_t, a_t, z_t$ and $r_t$ variables of $\tau$ in Section 3.1. The Policy Rewards corresponding to controller calls (marked in red) are the sum of the task rewards over the course of the next policy call—these are computed in the learner thread, since they depend on the future not known to the actor thread at the

current time step.

| State | $s_1$ | $s_1$ | $s_2$ | $s_3$ | $s_4$ | $s_5$ | $s_5$ | $s_6$ | $s_7$ | $s_7$ | ... |
|---|---|---|---|---|---|---|---|---|---|---|---|
| Action | $(\omega_1, l_2)$ | $a_1^{\text{env}}$ | $a_2^{\text{env}}$ | $a_3^{\text{env}}$ | $a_4^{\text{env}}$ | $(\omega_3, l_5)$ | $a_5^{\text{env}}$ | $a_6^{\text{env}}$ | $(\omega_2, l_4)$ | $a_8^{\text{env}}$ | ... |
| Task Reward | - | $r_1$ | $r_2$ | $r_3$ | $r_4$ | - | $r_5$ | $r_6$ | - | $r_8$ | ... |
| Policy Reward | $\sum_{t=1}^{1+l_2} r_t$ | $r_1^1$ | $r_2^1$ | $r_3^1$ | $r_4^1$ | $\sum_{t=5}^{6} r_t$ | $r_5^3$ | $r_6^3$ | $\sum_{t=8}^{8+l_4} r_t$ | $r_8^2$ | ... |
| Policy Index | $\Omega$ | $\omega_1$ | $\omega_1$ | $\omega_1$ | $\omega_1$ | $\Omega$ | $\omega_3$ | $\omega_3$ | $\Omega$ | $\omega_3$ | ... |
| Termination | 0 | 0 | 0 | 0 | 0 | 0 | 0 | 1 | 0 | 0 | ... |

*Table 9.* Example trajectory produced by the environment wrapper. Here $l_k = 2^k$ represents the option execution length output by the controller. The quantities in red represent the controller rewards, which are the sums of the task rewards over the course of execution of the option chosen by the controller. These are computed later in the learner thread (see above), and are included for completeness. Note that the second such sum is truncated to less than $l_5$ steps due to the episode termination.

### D.4. Parallelized V-trace

```
1
2        """
3        This function computes advantages and value targets for all policies in the batch
         simultaneously. The arguments are:
4
5
6        ratios: ratio of action probs between current and old policy
7        values: bootstrapped value predictions
8        dones: episode terminals
9        rewards: rewards of mixed type, see Policy Reward in Table 7.
10       rho_hat: V-trace truncation parameter
11       c_hat: V-trace truncation parameter
12       num_trajectories: number of trajectories in the batch
13       recurrence: number of timesteps in the batch
14       gamma: discounting factor
15       policy_indx: the Policy Index in Table 7, also z_t in Section 3.1
16       num_policies: total number of policies (options and controller, i.e. |\Omega| + 1).
17       """
18
19       def _compute_vtrace_sol(
20               ratios,
21               values,
22               dones,
23               rewards,
24               rho_hat,
25               c_hat,
26               num_trajectories,
27               recurrence,
28               gamma,
29               policy_indx,
30               num_policies,
31       ):
32           vtrace_rho = torch.min(rho_hat, ratios)
33           vtrace_c = torch.min(c_hat, ratios)
34
35           # tensors to store the advantages and value predictions
36           adv = torch.zeros((num_trajectories * recurrence,))
37           vs = torch.zeros((num_trajectories * recurrence,))
38
39
40           next_values = torch.zeros(num_trajectories, num_policies)
41           next_vs = torch.zeros(num_trajectories, num_policies)
42           delta_s = torch.zeros(num_trajectories, num_policies)
43
44           # V-trace returns are computed using a base case followed
45           # by recurrence relation. This marks which policies the
46           # base case  is handled for.
47           is_base_case_handled = torch.zeros(
48               num_trajectories, num_policies, dtype=torch.bool
49           )
50
51           # When an episode ends, we need to zero out the returns for
52           # each policy using the last timestep it is executed for
53           # before the episode ends.
54           is_episode_done = torch.zeros(
55               num_trajectories, num_policies, dtype=torch.bool
56           )
57
58           for i in reversed(range(recurrence)):
59               current_policies_one_hot = F.one_hot(
60                   policy_indx[i::recurrence], num_classes = num_policies
61               ).bool()
62
```

```
63          rewards = rewards[i::recurrence]
64          curr_dones = dones[i::recurrence].bool()
65
66          # when we encounter a "done", mark all policies as done.
67          # we will unmark the ones at the current timestep for
68          # which we mask out the returns.
69          is_episode_done = is_episode_done | curr_dones.view(-1, 1)
70
71          dones = is_episode_done[current_policies_one_hot].to(dtype)
72          not_done = 1.0 - dones
73          not_done_times_gamma = not_done * gamma
74
75          curr_values = values[i::recurrence]
76          curr_vtrace_rho = vtrace_rho[i::recurrence]
77          curr_vtrace_c = vtrace_c[i::recurrence]
78
79          # we have accounted for the latest episode termination
80          # of the current policies in 'not_done_times_gamma',
81          # so reset this until the next 'done' is encountered.
82          is_episode_done[current_policies_one_hot] = False
83
84
85          if i < recurrence - 3:
86              controller_indx = num_policies - 1
87              trajs_with_changed_options = (
88                  (policy_indx[(i+1)::recurrence] == controller_indx) &
89                  (policy_indx[i::recurrence] != policy_indx[(i+2)::recurrence])
90              )
91              # for any trajectories where the option switched,
92              # reset the base case so that bootstrapped returns
93              # are applied
94              is_base_case_handled[current_policies_one_hot] = \
95                  is_base_case_handled[current_policies_one_hot] & \
96                  ~trajs_with_changed_options
97
98
99          base_case_indices = (~is_base_case_handled) & current_policies_one_hot
100         base_case_indices_any = torch.any(base_case_indices, dim = 1)
101
102         next_values[base_case_indices] = (
103             values[i :: recurrence][base_case_indices_any]
104             - rewards[i :: recurrence][base_case_indices_any]
105         ) / gamma
106
107        next_vs[base_case_indices] = next_values[base_case_indices]
108
109         is_base_case_handled = is_base_case_handled | base_case_indices
110
111         if not is_base_case_handled.any().item():
112             continue
113
114         delta_s[current_policies_one_hot] = curr_vtrace_rho * (
115             rewards
116             + not_done_times_gamma * next_values[current_policies_one_hot]
117             - curr_values
118         )
119
120         adv[i::recurrence] = curr_vtrace_rho * (
121             rewards
122             + not_done_times_gamma * next_vs[current_policies_one_hot]
123             - curr_values
124         )
125
126         next_vs[current_policies_one_hot] = (
127             curr_values
```

```
128                    + delta_s[current_policies_one_hot]
129                    + not_done_times_gamma
130                    * curr_vtrace_c
131                    * (next_vs[current_policies_one_hot] -
132                    next_values[current_policies_one_hot])
133                )
134            vs[i::recurrence] = next_vs[current_policies_one_hot]
135            next_values[current_policies_one_hot] = curr_values
136
137      return adv, vs
```

# E. Additional Experiment Results

## E.1. Additional NetHack Characters

Here we report results with additional NetHack characters. Most prior work (Klissarov et al., 2024; 2025b; Zheng et al., 2024) uses the Monk character, however this is only one out of 13 characters in the game. Here we compare all methods on two other characters: the Ranger and Archaeologist. The trends we observed for the Monk are repeated here: SOL and SOL+Motif significantly outperform the other methods, and their performance continues to improve over the course of 30 billion training samples. This shows that our conclusions are not particular to the Monk character.

We also note that the scores for the Ranger and Archaeologist are significantly lower than the Monk, which is likely due to the fact that the Monk starts proficient in unarmed combat and can succeed in the early game without needing to learn how to equip weapons and armor.

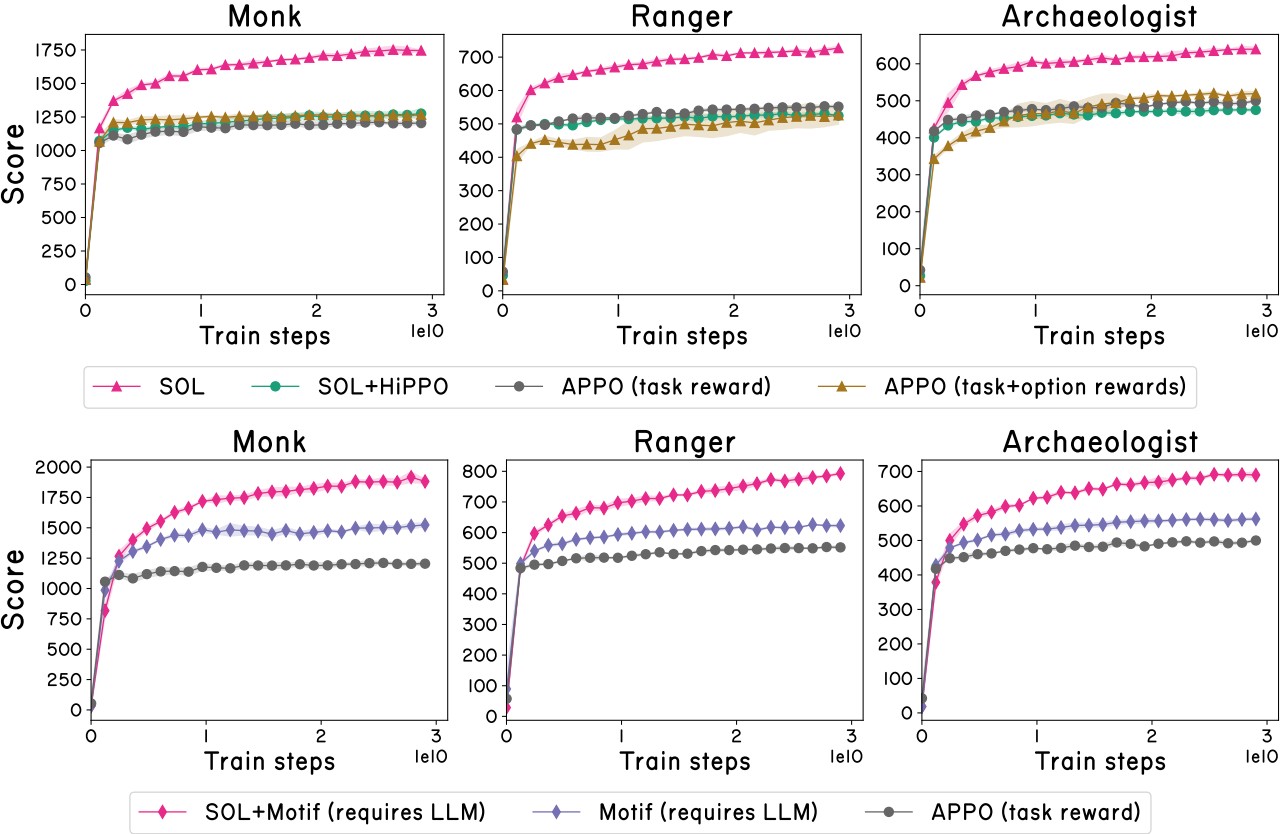

*Figure 9.* Results on `NetHackScore` for three different characters. Curves represent the mean and shaded regions represent two standard errors computed over 5 seeds.

### E.2. Visualizations and Analysis

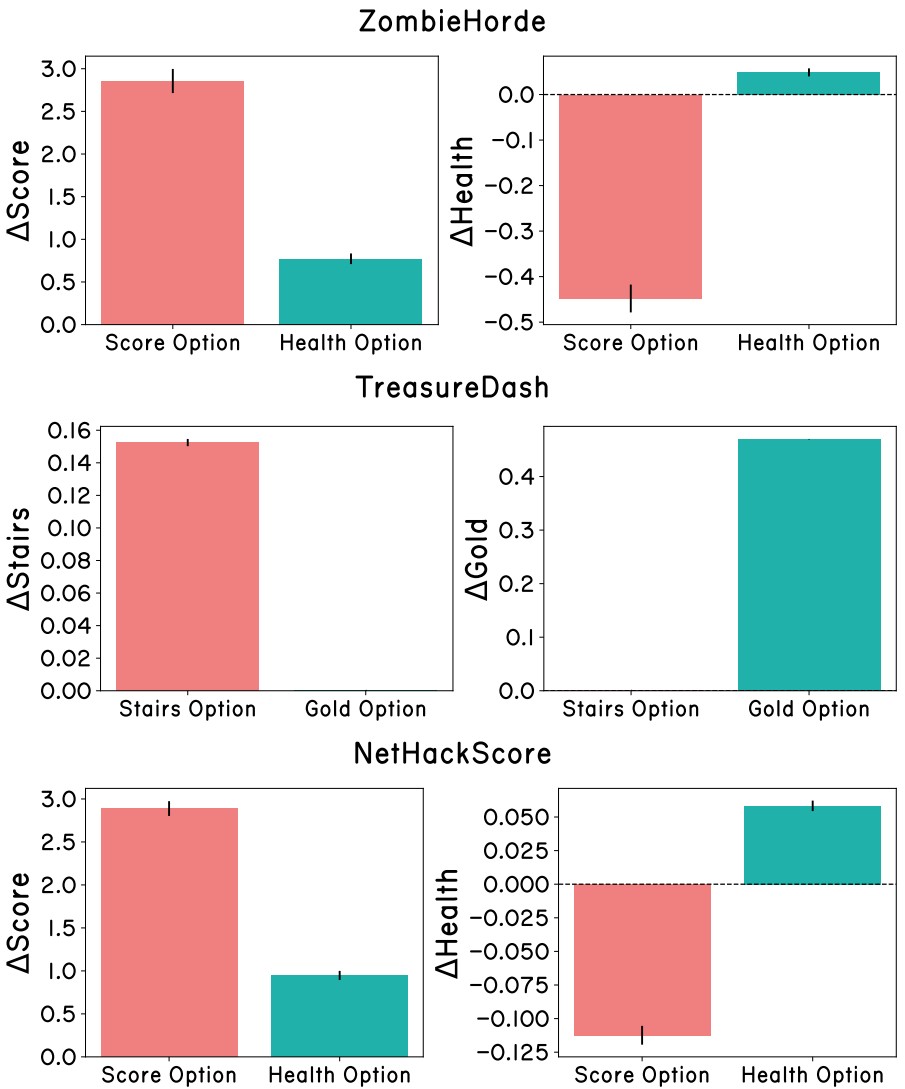

*Figure 10.* Option mean returns, normalized by option execution length. Error bars represent two standard errors computed over 500 episodes.

In this section we provide visualizations which help shed light on SOL's behavior. In Figure 10, for each environment we report the average return in terms of each option reward when executing each option policy. On both ZombieHorde and NetHackScore, the `Score` option accumulates higher score than the `Health` option (as shown by its higher $\Delta$`Score` return), but sustains damage over time (as shown by its negative $\Delta$`Health` return). The `Health` option accumulates less score, but *recovers* health over time (as shown by its positive value in terms of $\Delta$`Health`, enabling the agent to survive longer overall. For TreasureDash, the `Stairs` option achieves positive $\Delta$`Stairs` reward (indicating it has descended a staircase) and no $\Delta$`Gold` reward (indicating it has collected no gold), whereas the `Gold` reward is the opposite. Overall, this shows that SOL is able to learn different options which produce distinct behaviors.

We additionally measured overlap between option policies by computing the normalized mutual information (NMI) between their action distributions over the course of 100 evaluation episodes, shown in Table 10. The NMI between variables $X$ and $Y$ is defined as: $I(X,Y)/\min(H(X),H(Y))$ and can range between 0 (fully independent) and 1 (fully redundant). We see that the NMI is low for all three environments, providing further evidence that the sub-policies learn distinct behaviors. Interestingly, it is lowest for TreasureDash, which also has the most distinct options in terms of return, with no overlap between their respective rewards.

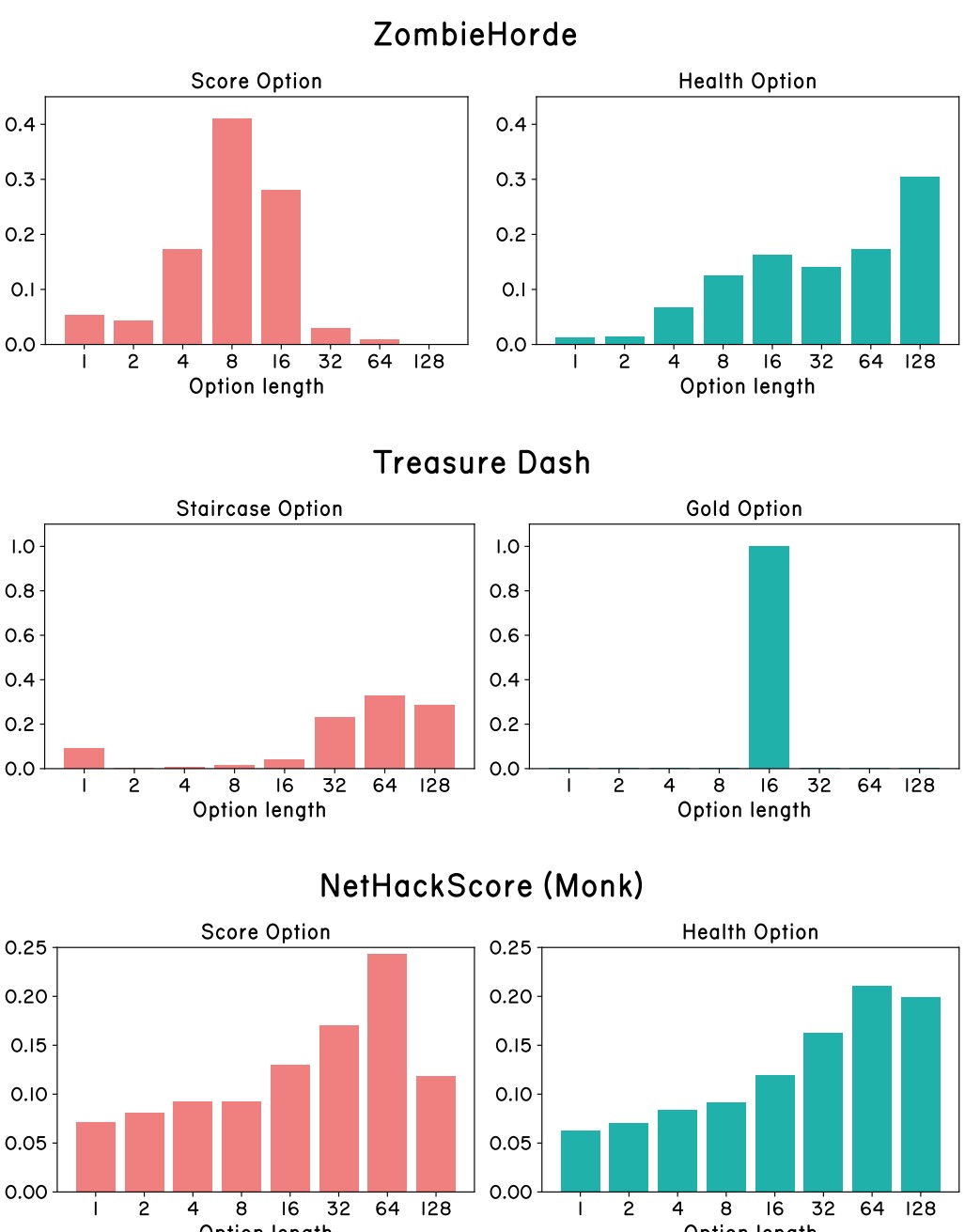

*Figure 11.* Distributions of option execution lengths chosen by the controller. The distribution is non-uniform, which indicates learning on the part of the controller. Longer option lengths are chosen more frequently than short ones. The Score option tends to be executed for the longest option length (128 steps) less often than Health. This may be because calling Score for longer than is optimal carries a higher risk than for Health: executing Score for too long may result in too much combat and agent death, whereas executing Health for too long will result in the agent wasting turns trying to heal at full health, which has fewer negative consequences. Distributions are computed over 500 test episodes.

| Environment | Normalized Mutual Information |
|---|---|
| ZombieHorde | 0.154 |
| TreasureDash | 0.018 |
| NetHack | 0.388 |

*Table 10.* Normalized Mutual Information (NMI) between action distributions of different option policies, computed over 100 episodes. The NMI is defined between 0 and 1.

In Figure 11, for each environment we plot the distributions of option lengths selected by the controller for each option. On ZombieHorde, the Score option tends to be called for shorter lengths than Health. This may be explained by the fact that healing takes a long time, around 10 time steps per hit point: at experience level 1, healing from 7/14 hit points back to full health takes ~70 time steps. Also, executing the Score option involves fairly high uncertainty due to the stochasticity of NetHack's combat system, where damage is dealt randomly based on various statistics: it may be that the agent gets lucky defeats several monsters in a row, or it may be unlucky and sustain high damage at the beginning, in which case it needs to switch back to the Health option. Choosing shorter option lengths for Score allows the agent to switch back to the Health option more quickly if needed. In contrast, healing is mostly deterministic and there is less downside to selecting the Health option for longer than needed. In TreasureDash, the controller very precisely chooses the optimal execution length of 16 for the Gold option (the optimal policy moves right for 16 steps to get 8 gold, then moves left for 24 steps to the stairs), and assigns similar lengths to any of the 3 optimal lengths for the Staircase option (32, 64, 128). For NetHackScore, the option lengths are more spread out, although Score is still skewed somewhat shorter than Health. We note that healing is shorter in NetHackScore, because the action space includes extended movement actions (such as MOVEFAR) than take several game turns, and executing one of these speeds up healing from the perspective of the agent—this may explain why the difference in option lengths is less pronounced than for ZombieHorde, even though the option rewards are the same for both environments.

In Figure 12, we plot the fraction of controller calls to each option conditioned on the agent's health and experience level. The controller calls the Health option more frequently when the agent's health as low, which makes sense since this enables the agent to recover its health and survive longer. Interestingly, the controller also tends to call the Health option more often at low experience levels (96% of the time at Experience Level 1). Upon visualizing trajectories, we found that the agent still fights monsters that attack it when executing the Health option, but does not seek them out. This results in the agent staying at the first few dungeon levels, fighting weaker monsters that appear, and gaining some experience levels. It then begins calling the Score option more frequently, resulting in it attacking monsters and exploring further into the dungeon. This is similar to successful human gameplay, which requires careful progression of dungeon levels only when the agent is strong enough (contributors).

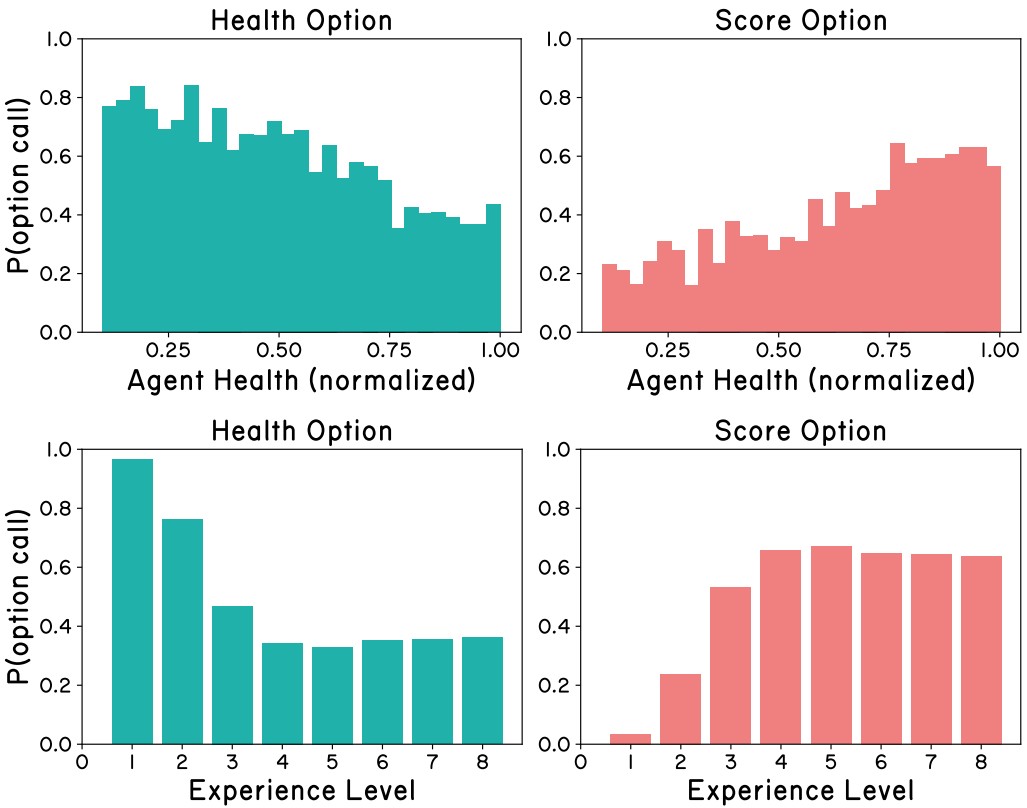

*Figure 12.* Fraction of controller calls to the `Health` and `Score` options for Monk, conditioned on the agent's normalized health (current hit points divided by maximum hit points) and experience level. The controller calls the `Health` option more frequently at low health, enabling the agent to recover and survive longer, and at low experience levels, when the agent is still weak. Distributions are computed over 500 test episodes.

## E.3. NetHack Bootstrapping and Option Length Ablation

We report results for SOL on NetHack studying the effect of removing the adaptive option length mechanism and value function bootstrapping on option switches described in Section 3.1. Figure 13 compares three variants: with bootstrapping and adaptive option lengths, without bootstrapping but with adaptive option lengths, and without bootstrapping and with a fixed option length of 16 (we previously tuned the option length during development and found that 16 gave the best results). We see that performance progressively degrades as each component is removed.

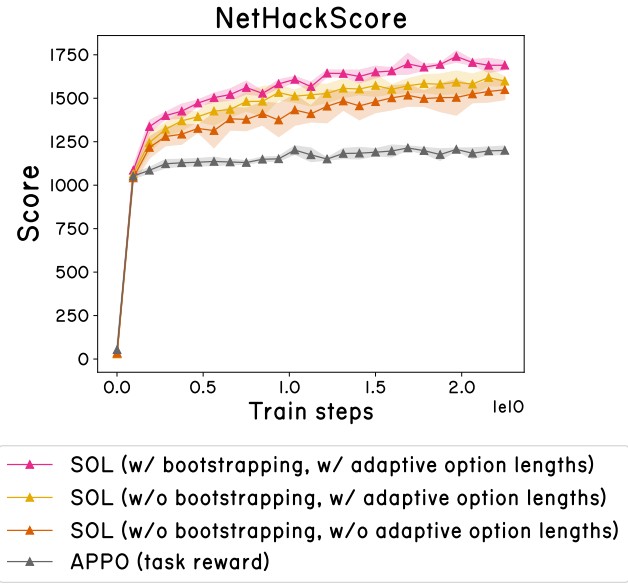

*Figure 13.* Ablation results on NetHackScore. Shading represents standard errors over 5 seeds.

## E.4. MiniHack Option Length Ablations

**Fixed vs. Adaptive Option Lengths** In Figure 14 we report the final results for both MiniHack environments when using different fixed option lengths in $\{2, 4, 8, 16, 32, 64\}$. In this setting, every time the controller selects an option it is always executed for the same fixed number of steps. Having fixed lengths which are either too long or too short lengths hurts performance. In contrast, our adaptive selection mechanism is able to automatically tune the option lengths, and performs comparably to the best fixed option length.

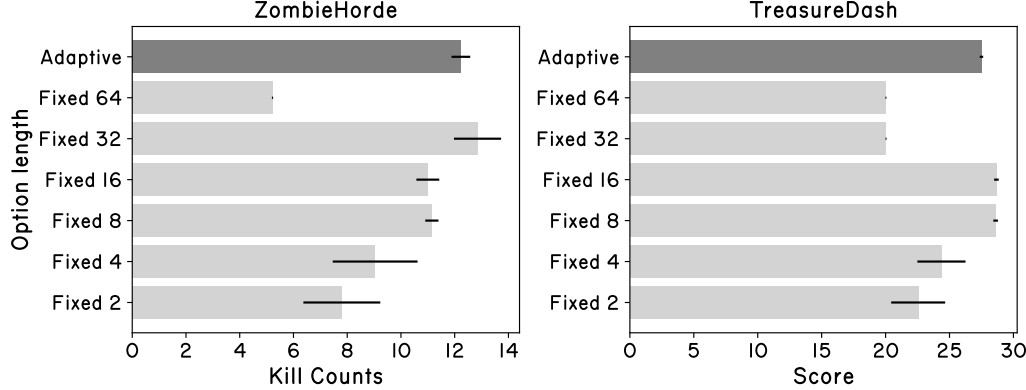

*Figure 14.* Final performance on ZombieHorde and TreasureDash for different fixed options lengths as well as the adaptive option lengths. Bars represent standard errors over 5 seeds.

**Number of Option Lengths**    In Figure 15 we study how the performance of SOL (with adaptive option lengths) changes as we vary the number of option lengths the controller can choose from. Recall that each time the controller is called, it selects an option execution length from the set $L = \{2^i : 1 \leq i \leq M\}$, with $M = 8$ our default value in all experiments. Here we compare values of $M$ in $\{4, 8, 12, 16\}$.

On ZombieHorde, compared to $M = 8$ there is a drop in performance for other values of $M$, however all values of $M$ still outperform the flat baseline. On TreasureDash, there is no difference in terms of final performance and we only observe slower convergence for $M = 4$. These results suggest that SOL exhibits some sensitivity to the $M$ hyperparameter, but one can still see improvements over a flat agent even if it is not set optimally.

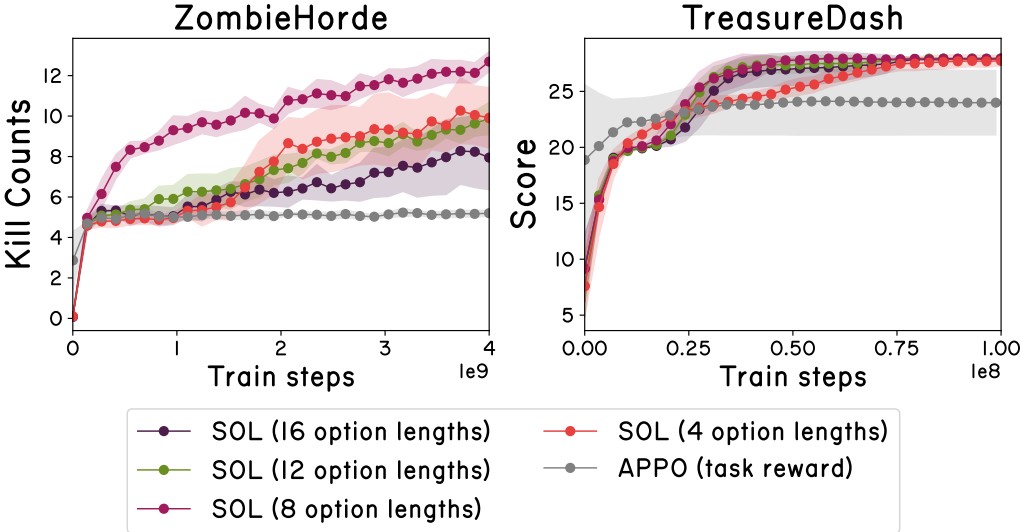

*Figure 15.* Performance of SOL with different numbers of option lengths for the controller to choose from. Shaded region represents two standard errors computed over 10 seeds.

## E.5. MiniHack Option Reward Scaling Ablation

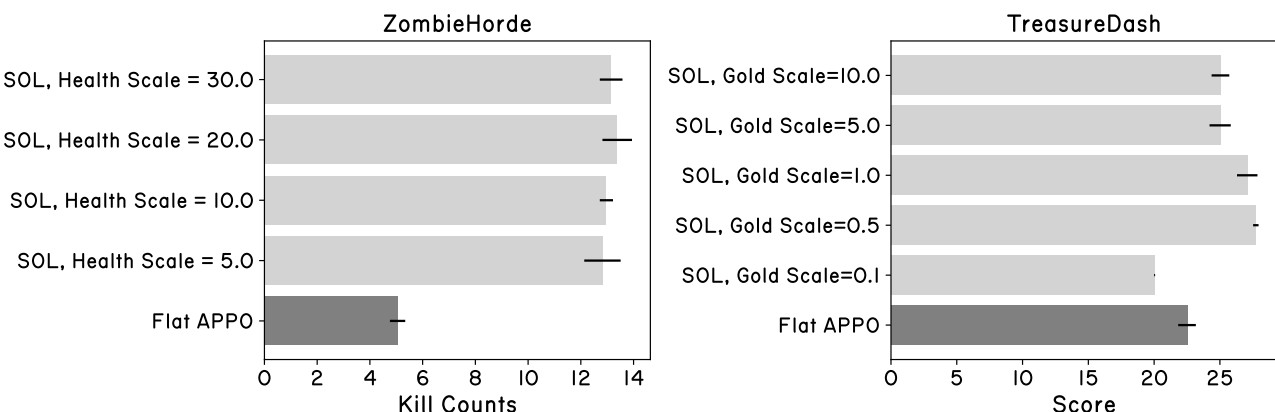

*Figure 16.* Final performance on ZombieHorde and TreasureDash for different scaling coefficients of the non-task-reward option. Bars represent standard errors over 5 seeds.

## E.6. MiniHack Option Quality Ablation

In Figure 17 we study how the performance of SOL changes in the presence of redundant or useless options on both MiniHack tasks. We compare the following variants:

- SOL : our default version, which has two options that are both useful for the task ($\Omega = \{$`Score, Health`$\}$ for ZombieHorde, $\Omega = \{$`Stairs, Gold`$\}$ for TreasureDash).

- `SOL(+2 duplicate options)`: has both original options duplicated once each. Its option set is $\Omega = \{$`Score, Score2, Health, Health2`$\}$ for ZombieHorde and $\Omega = \{$`Stairs, Stairs2, Gold, Gold2`$\}$ for TreasureDash. Here `Score2` is an option with identical reward as `Score`, and same for the other options.

- `SOL(+8 duplicate options)`: has both original options duplicated 4 times each. Its option set is $\Omega = \{$`Score,...,Score5, Health,...,Health5`$\}$ for ZombieHorde and $\Omega = \{$`Stairs,...,Stairs5, Gold,...,Gold5`$\}$ for TreasureDash.

- `SOL(+2 useless options)`: has 2 options added which are unrelated to the task at hand. For ZombieHorde, the option set is $\Omega = \{$`Score, Health, Gold, Scout`$\}$ and for TreasureDash the option set is $\Omega = \{$`Stairs, Gold, Scout, Health`$\}$. Here `Scout` is a reward measuring exploration taken from (Küttler et al., 2020).

Results are shown in Figure 17. Adding duplicates of options that are useful for the task at hand does not significantly change performance. Adding useless options (which are unrelated to the task at hand) slows down learning on both tasks, which is unsurprising: without prior knowledge, the agent must learn through experience which options are useful and which are not (also recall that we have an entropy bonus on the controller which encourages it to sample all options with some probability). However, on both tasks the agent with useless options is able to eventually match the performance of the others, given sufficient training.

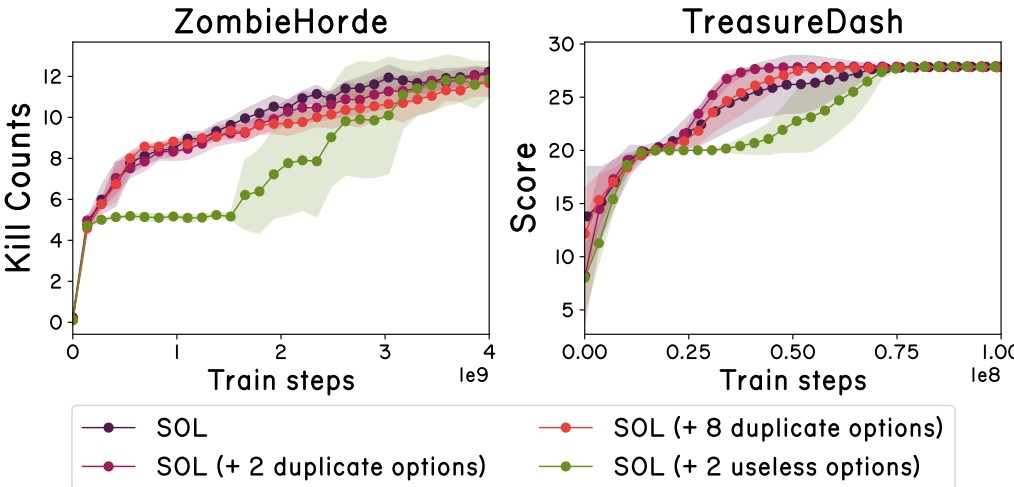

*Figure 17.* Performance of SOL with duplicate or useless options added. Shaded region represents two standard errors computed over 5 seeds.

### E.7. MiniHack Sample Complexity Ablation

SOL inherits PPO's ability to do multiple off-policy updates on the same batch of data. Off-policy updates can improve sample efficiency and/or increase GPU utilization when the environment throughput is lower, but can also cause instability due to the mismatch between the updated policy and the behavior policy used to generate the data, which PPO's clipping thresholds attempt to remedy. In Figure 18 we study how performance changes as we vary the number of off-policy PPO epochs on both MiniHack tasks. On ZombieHorde, we see a significant improvement in sample efficiency as we increase the number of epochs, with no loss of performance. On TreasureDash, there is a little improvement and even some deterioration in performance as we increase the number of epochs. It is possible that further tuning the PPO clipping parameters could improve performance when the number of epochs is greater than one. Overall, these experiments illustrate that SOL faces similar trade-offs as PPO concerning the number of off-policy epochs: they can improve sample complexity in some settings, but may introduce instability in others.

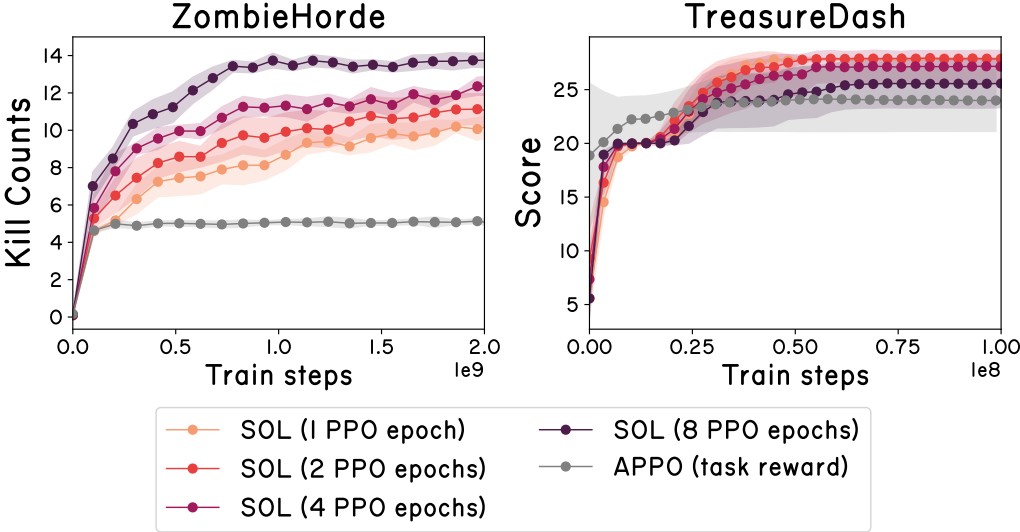

*Figure 18.* Performance of SOL with varying numbers of PPO epochs. Shaded region represents two standard errors computed over 10 seeds.

## E.8. Flat APPO results on PointMaze

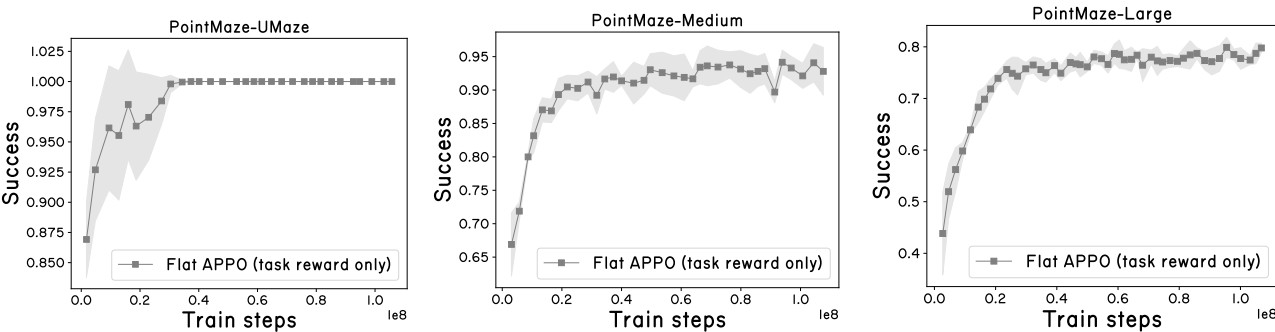

*Figure 19.* Flat APPO agents trained on default PointMaze environments from Gymnasium Robotics (de Lazcano et al., 2024) are largely able to solve all PointMaze environments, indicating that hierarchy is not needed for these maze layouts.

