# OpenReview forum: "Scalable Option Learning in High-Throughput Environments"
_ICML.cc/2026/Conference — ICML 2026 spotlight_

### Official Review · Reviewer_jG1P · 2026-02-15

**Soundness:** 3
**Presentation:** 3
**Significance:** 3
**Originality:** 2
**Overall Recommendation:** 5
**Confidence:** 4

**Summary:**

This paper introduces a hierarchical reinforcement learning (HRL) algorithm, named Scalable Option Learning (SOL), that learns both the low-level options and the high-level controller simultaneously using a single neural network. The method is designed to be highly parallelizable and to scale to high-throughput RL domains, where agents typically collect billions of experience samples. The authors compare SOL in challenging MiniHack and NetHack domains, as well as in continuous control problems (Mujoco). The empirical results indicate that SOL is able to significantly outperform previous HRL methods both in compute time, sample efficiency and asymptotic performance in more challenging domains.

**Compliance With Llm Reviewing Policy:**

Affirmed.

**Final Justification:**

The rebuttall address my main concerns. Overall, I believe this is a solid paper with a significant empirical contribution.

**Key Questions For Authors:**

- Suppose there is an optimal controller that selects an option for 32 time steps. Given the proposed objective, nothing guarantees that a controller that selects this same option 16 times with option length equal to 2 would be learned instead. This problem in known in the HRL literature as “option degeneracy” (see [2]). Did the authors observed this phenomenon?

- As the options are being updated, the experiences used to train the controller become invalid. This is a non-stationary learning process. How is this handled by the proposed method? I suppose experiences cannot stay in the buffer for long to train the controller since they becamo inconsistent as the options change.

- The paper could benefit is terms of clarity by the addition of a pseudocode of the entire algorithm.

- In many of the experiments, the controller reward is the same as the reward one of the options. Did the authors have any observations regarding the relevance of the level of alignment between the multiple reward functions?

[2] Luo, Z., Zhang, Y., and Wang, Z. (2023). Does Hierarchical Reinforcement Learning Outperform Standard Reinforcement Learning in Goal-Oriented Environments? In Goal-Conditioned Reinforcement Learning Workshop, Neural Information Processing Systems.

**Limitations:**

The authors discussed a few of the limitations of the proposed method. However, there are still some points that warrant discussion, such as the non-stationarity of the loss function and the lack of performance guarantees.

**Strengths And Weaknesses:**

### Strengths
- From an engineering point of view, the proposed architecture and system design, which allows for and end-to-end and parallized implementation of the algorithm, is a significant contribution to the RL community and may have good impact.
- The empirical results are impressive and considered complex benchmarks tailored for HRL.
- The proposed method significantly outperforms previous HRL methods both in running time and sample efficiency.

### Weaknesses

- From a theoretical and algorithmic point of view, there is not much novelty/originality in the proposed algorithm. The controller and options are trained by optimizing separate hand-designed reward functions, and the problem of defining initiation/termination sets for options is addressed by adding the option length as part of the action space of the controller.
- There is no discussion or results regarding the theoretical guarantees of the proposed method. For instance it is unclear how much optimality may be lost due to the approximation in line 162.
- The proposed joint optimization of the value and policy losses for the controller and options result in non-stationary behavior, due to the fact that every time that the options change, the value function and policy of the controller become outdated. This problem is widely known in the HRL literature [1]. I would expect at least a discussion on how/why this problem does not affect the proposed method.


[1] “Discovering Temporal Structure: An Overview of Hierarchical Reinforcement Learning”. Klissarov et al., 2025. https://arxiv.org/pdf/2506.14045

---

> ### Author Rebuttal · Authors · 2026-03-30
>
> Thank you for your time spent reviewing our paper and for the overall positive review. Please see our responses to your comments and questions below.
>
> >There is no discussion or results regarding the theoretical guarantees of the proposed method. [...]
>
> The focus of this paper is on designing a scalable and empirically effective HRL algorithm. However, although flat policy gradient algorithms are known to converge to local optima, such an analysis becomes more difficult in our setting because of the dependence between the controller and option policies. Note that prior analyses such as [1,2] do not apply in our case, because they optimize a single reward, whereas we in our case, the option policies and controller optimize different rewards. We believe analyzing this type of algorithm would be interesting future work.
>
> >The proposed joint optimization of the value and policy losses for the controller and options result in non-stationary behavior, [...]. I would expect at least a discussion on how/why this problem does not affect the proposed method. […] As the options are being updated, the experiences used to train the controller become invalid. [...] I suppose experiences cannot stay in the buffer for long to train the controller since they becamo inconsistent as the options change.
>
> We recognize that non-stationarity exists, however we empirically found that our system worked well despite it. We believe this may be for two reasons. First, note that each option policy optimizes its reward independently of the other option policies as well as the controller, thanks to the value objective described in section 3.1. The only dependence is in the start state distributions the options are initiated from – the value estimates are unaffected by the other policies. In other words, for some option $\omega$, state $s$ and action $a$, the advantage $A^\omega(s, a)$ and value $V^\omega(s)$ do not depend on what the controller or other option policies do after $\omega$ is finished executing, and the only influence they have on $\omega$ is in how often $s$ is visited in the first place.
>
> Second, since our method uses an on-policy algorithm (PPO), we do not maintain a replay buffer that gets stale as the option policies change. This is unlike methods such as HIRO [3], that use a replay buffer and need to include off-policy corrections to account for the fact that a given option policy executed at a given state far in the past will induce different controller rewards if it were executed at that same state at the present time.
>
> We will include a discussion of this in the updated paper, thanks for bringing it up.
>
>
> > Suppose there is an optimal controller that selects an option for 32 time steps. Given the proposed objective, nothing guarantees that a controller that selects this same option 16 times with option length equal to 2 would be learned instead. This problem in known in the HRL literature as “option degeneracy” (see [2]). Did the authors observed this phenomenon?
>
> If the controller does not have a discount factor, then both of these choices would give the same expected return and the controller will prefer them both equally. However, the first choice may be easier to _discover_ than the second, because the chance of selecting it through random exploration is: `(1/num_options)*(1/num_option_lengths)` whereas the second is: `((1/num_options)*(1/num_option_lengths))**16`, which is much smaller.
>
> > The paper could benefit in terms of clarity by the addition of a pseudocode of the entire algorithm.
>
> We will add a pseudocode overview of SOL in the Appendix. Also, we are currently working on an implementation of SOL based on the cleanRL single-file implementation of PPO, and will include this in our code release. This should help make the algorithm easy to understand while also providing an implementation that is easy to modify.
>
> > In many of the experiments, the controller reward is the same as the reward one of the options. Did the authors have any observations regarding the relevance of the level of alignment between the multiple reward functions?
>
> We are not completely sure what the reviewer means by alignment here. However, we found that including the task reward as one of the option rewards was convenient, because the controller can easily match the behavior of the flat agent by repeatedly selecting this option, but it can also select different options which induce different behavior.
>
>
> We hope that our clarifications and paper updates address your comments, and that you will consider raising your score from Weak Accept to Accept. If you have remaining concerns that prevent you from doing so, please let us know and we will be happy to discuss further.
>
>
> [1] The Option Critic Architecture, Bacon et al. (AAAI 2017).
>
> [2] Learning Abstract Options, Riemer et al. (NeurIPS 2018).
>
> [3] Data-Efficient Hierarchical Reinforcement Learning, Nachum et al. (NeurIPS 2018)

---

> > ### Author Rebuttal · Reviewer_jG1P · 2026-04-02
> >
> > I thank the authors for the detailed response. I strongly suggest they include in the revised version of the paper a discussion on: 1) non-stationarity, and 2) option degeneracy.
> >
> > Regarding point 2), I agree with the authors that the option with extended length would be easier to discover. However, there always exists a controller that uses only primitive actions (option length equals 1) that is optimal. If, by increasing the option length, an agent would incur only very small performance losses, the agent would always prefer to use primitive actions.

---

### Official Review · Reviewer_MZgW · 2026-02-27

**Soundness:** 4
**Presentation:** 4
**Significance:** 4
**Originality:** 4
**Overall Recommendation:** 6
**Confidence:** 4

**Summary:**

The paper proposes Scalable Option Learning (SOL), a hierarchical RL method designed to make options work in high-throughput training setups. The core idea is to allow multiple option policies and the controller to be trained efficiently within a single shared network, enabling GPU-friendly batching and asynchronous training. This makes hierarchical agents almost as fast as flat ones while being dramatically faster than prior hierarchical methods. SOL performs well on MiniHack and PointMaze, and achieves state-of-the-art results on NetHackScore, while also being compatible with learned or LLM-generated intrinsic rewards.

**Compliance With Llm Reviewing Policy:**

Affirmed.

**Final Justification:**

The rebuttal addressed all of my questions and remarks. I find this work very impressive (see the strengths above) and recommend a spotlight, if not oral.

**Key Questions For Authors:**

1. How to select $L$? Is some domain/environment-specific knowledge always required? What happens to SOL if $L$ contains too few or too many option length values?
2. How tightly coupled is SOL with Sample Factory? From my experience, while Sample Factory provides excellent throughput, it is not particularly flexible when it comes to anything customized. How much effort would be required to extract the core SOL components and integrate them into some other high-performance RL library?
3. What happens to the active option when an episode terminates?

**Limitations:**

The limitations are discussed in Appendix A. I don't have much to add on top of that, apart from potentially what I've pointed out in the weaknesses above.

**Strengths And Weaknesses:**

# Strengths

1. Although the core contribution is not an algorithmic advancement in the traditional sense, but rather engineering-oriented, I nevertheless see a lot of value in this direction. Several sub-domains of RL are notoriously bottlenecked by computation and the lack of efficient parallelization. While there are quite a few recent developments on fast environments and efficient benchmarks, there is less attention being paid to building high-throughput training pipelines for more involved non-trivial scenarios, such as hierarchical RL. In that sense, I believe SOL has clear practical significance and impact.
2. On a more practical side, the design choices are well thought out and clearly motivated. A single network, conditioned on a policy index, for the options and the controller, is a clever way to maintain uniform shapes. Masking returns in the learner helps avoid splitting trajectories by option and running separate backward passes. SOL can incorporate different sources of option rewards, whether hand-designed, LLM-generated, or diversity-based objectives from skill discovery. Altogether, the 35x-580x speedup makes the hierarchical RL approach a lot more viable.
3. The evaluation is convincing. The MiniHack tasks (ZombieHorde and TreasureDash) provide a simple and interpretable setting. We can directly observe how the agent balances the manually chosen option rewards to obtain the highest score, and how other baselines fail to do so. MuJoCo maze environments are often used to evaluate skill discovery and unsupervised RL, so the PointMaze serves as a suitable complementary continuous control setting. And most importantly, NetHackScore, a rich and notoriously complex, long-horizon environment, SOL reaches SOTA performance among RL agents. The comparisons include both a hierarchical baselines as well as an LLM-based method, and notably, combining SOL with an LLM-based reward generator yields even higher scores, and beats the standalone approach.
4. The writing, motivation, structure, and overall narrative of the paper are good. I found it straightforward to follow (1) how SOL compares to relevant existing methods, (2) why hierarchical RL is implementation-wise difficult to parallelize, (3) what the architecture, objective and training pipeline of SOL looks like, (4) the relevance of the selected evaluation environments for hierarchical RL, and (5) the discussion of the results. The code snippets and schematics (Figure 1) are useful for understanding how SOL functions under the hood.

# Weaknesses

1. According to Table 9, the wrapper seems to continue executing option $\omega_3$ after the episode terminates at $s_6$. This seems at odds with the wrapper pseudocode in Appendix D.3, where `reset()` sets `current_policy = "controller"`. Assuming the gymnasium convention is used, the reset is guaranteed to occur after termination. This would imply that the controller re-selects an option. Although Appendix D.4 shows that returns are properly zeroed out at episode boundaries, it remains unclear whether active options are reset on termination or whether they can “leak” to new episodes.
2. In most of the evaluation we only see 2 options, with the exception of 5 in PointMaze. Although a low number is common in prior work, the paper emphasizes scalability. I am curious if these computational gains can extend beyond high throughput, and enable scaling along the “hierarchical dimension” itself, i.e., to a larger $|\Omega|$. Appendix E.6 already investigates duplicate and useless options, but what if there are actually more useful options? This seems especially relevant for the DIAYN-combined setting, where skill discovery is often studied with dozens of skills.
3. It is not clear how sensitive SOL is to the choice of the option length set $L$. First of all, this has to be user-defined. The appropriate values for $L$ likely depend on the selection of $|\Omega|$,  but also the underlying environment dynamics and episode length. Since the controller selects $l \in L$, this sets the scene for how much temporal abstraction is available to the agent.
4. Given that this work is largely engineering-driven, access to the codebase would help in assessing the contribution. While there are several useful code snippets and implementation details in the paper, the authors do not provide the code as supplementary material or link to an anonymous repository.

### Minor Points

1. It could be useful to explicitly indicate that the schematic grids in Figure 1 represent tensors of shape $[B×T]$. It wasn’t immediately obvious how to interpret those grids.
2. In Figure 3 (right), the “(w/ LLM)” label suggests that all methods shown in that plot make use of LLM-based rewards. However, I don’t think APPO (task reward) makes use of any such thing and is likely included just as a soft lower bound. It would be helpful to clarify this (e.g., with an asterisk or short note) to avoid confusion.

---

> ### Author Rebuttal · Authors · 2026-03-30
>
> Thank you for your time spent reviewing our paper and for the positive review. Please see our responses to your comments and questions below.
>
> > According to Table 9, the wrapper seems to continue executing option $\omega$ after the episode terminates. This seems at odds with the wrapper pseudocode in Appendix D.3 [...] What happens to the active option when an episode terminates?
>
> Good catch! This was a typo in Table 9. You are correct that after the episode terminates, the controller is executed at the first observation of the next episode following the reset and a new option is selected. We will use the corrected table in the updated paper.
>
> > In most of the evaluation we only see 2 options, with the exception of 5 in PointMaze. Although a low number is common in prior work, [...] I am curious if these computational gains can extend beyond high throughput, and enable scaling along the “hierarchical dimension” itself [...]. What if there are actually more useful options? This seems especially relevant for the DIAYN-combined setting, where skill discovery is often studied with dozens of skills.
>
>
> This is a great question and is something we hope to investigate in future work. We did some preliminary experiments with more than 2 options on NetHack (such as eating food, gaining experience, etc), but the current combination with 2 options worked best. We note that adding extra options has a negligible effect on the computational throughout, since each extra option only increases the size of the controller action space and the policy embedding layer by 1. We believe the challenge is more one of finding good option rewards, or possibly parameterizing the options in a different manner - our framework should enable future work in this area at scale.
>
>
> > It is not clear how sensitive SOL is to the choice of the option length set . [...] The appropriate values for L likely depend on the selection of $|\Omega|$ , but also the underlying environment dynamics and episode length. […] How to select L? Is some domain/environment-specific knowledge always required? What happens to SOL if L contains too few or too many option length values?
>
> The option lengths are exponentially spaced, i.e. L={1, 2, 4, 8, …, 2**M}, hence they can cover a wide range of values. We used M=8 in all our experiments. We have run a new ablation on both MiniHack tasks where we sweep M over values in {4, 8, 12, 16}. On TreasureDash, there is no significant difference in terms of convergence speed or final performance. On ZombieHorde, compared to M=8 there is a modest drop in performance for M=4,12 and a bigger drop for M=16. However, all values of M still outperform the flat baseline. This suggests some sensitivity to this hyperparameter, but one can still see improvements over a flat agent even if it is not set optimally.
>
>
> >Given that this work is largely engineering-driven, access to the codebase would help in assessing the contribution. While there are several useful code snippets and implementation details in the paper, the authors do not provide the code as supplementary material or link to an anonymous repository.
>
> We definitely plan on sharing the full codebase if the paper is accepted. We agree that sharing the full code greatly helps reproducibility.
>
>
>
>
> >It could be useful to explicitly indicate that the schematic grids in Figure 1 represent tensors of shape B x T. It wasn’t immediately obvious how to interpret those grids.
>
> Thanks for the suggestion, that is a good idea. We will add these labels to Figure 1 in the updated paper.
>
> >In Figure 3 (right), the “(w/ LLM)” label suggests that all methods shown in that plot make use of LLM-based rewards. However, I don’t think APPO (task reward) makes use of any such thing and is likely included just as a soft lower bound. It would be helpful to clarify this (e.g., with an asterisk or short note) to avoid confusion.
>
> That is correct, APPO (task reward) does not use an LLM, and is included as a reference point to put the other results in context. We will clarify this in the figure.
>
> >How tightly coupled is SOL with Sample Factory? From my experience, while Sample Factory provides excellent throughput, it is not particularly flexible when it comes to anything customized. How much effort would be required to extract the core SOL components and integrate them into some other high-performance RL library?
>
> Our current implementation uses Sample Factory, but the design decisions are general and could be used with any other library. We are currently working on an implementation based on cleanRL, which tends to be less fast than Sample Factory but easier to read and modify. We will release this together with our Sample Factory implementation.
>
> We hope that our clarifications and paper updates address your questions and comments. If you have any remaining questions, we will be happy to discuss further.

---

> > ### Author Rebuttal · Reviewer_MZgW · 2026-04-03
> >
> > Thank you for responding to each of my remarks. I consider all the responses satisfactory. I find your work very impressive and will raise the score.

---

### Official Review · Reviewer_eAtn · 2026-03-10

**Soundness:** 4
**Presentation:** 4
**Significance:** 3
**Originality:** 3
**Overall Recommendation:** 5
**Confidence:** 4

**Summary:**

This paper proposes Scalable Option Learning(SOL), a hierarchical reinforcement learning algorithm designed for high-throughput environments. The proposed method are validated in environments such as MiniHack, NetHack and Mujoco PointMaze, achieving a ~35x to 580x higher throughput than existing hierarchical RL implementations, while retaining 86% of the speed of a flat APPO baseline. SOL outperforms both flat APPO and HiPPO in final task performance.
Two methods are introduced in this paper, for hierarchical RL at scale i) a novel bootstrapping objective for option value functions, enabling independent optimization from mixed behavior trajectories; and ii) system-level optimization for hierarchical RL, which allows option policies being independently optimized with trajectories of mixed policies and, allows higher throughput for hierarchical RL, respectively. On system level, this paper i) integrates controller and option policies in a single neural network, ensuring that each control or action decision step is finished with one forward pass, ii) defining environment wrapper for hierarchical RL, environment does not update during controller step, and iii) efficient parallelized masking when computing the advantages and value targets for each policy.

**Compliance With Llm Reviewing Policy:**

Affirmed.

**Final Justification:**

The original review involves two main concerns, theoretical analysis and soundness.

The authors have addressed the issues about theoretical analysis by pointing out that SOL optimizes different rewards, resulting in rather complex optimization dynamics, which may shift the discussion to a different topic than the one this paper was trying to address.

It could be seen from the paper that the authors have commitment in reproducibility of their work. They also addressed main concerns in the previous review by conducting and presenting more ablation experiments, reinforcing my previous assessment of soundness.

**Key Questions For Authors:**

To my knowledge, environments like NLE consumes minimal CPU resources, that is also, as I understand it, how your research can parallelize at scale. How is the compatibility of your work with other environments than gym Robotics? Could it also achieve a better throughput performance?
It's noted in the paper that "efficient parallelized masking when computing the advantages and value targets for each policy". Is there a particular reason that you mentioned the concept "efficient masking"?
Have you tried to compare the scale of neural network between a flat and hierarchical architecture? Could the difference in performance be related with this factor?

**Limitations:**

Yes

**Strengths And Weaknesses:**

Soundness
Strengths:
The implementation is backed with codes and system design graphs, demonstrating engineering rigor. Key solutions are well-motivated and implemented.
The experiments on multiple environments with different characteristics provide strong evidence for SOL's general applicability.
This paper provides thorough ablation analyses on both theoretical and system level designs.
This paper provides detailed hyperparameters, architecture descriptions and even code for the parallelized V-trace computation, making reproduction feasible.
Weakness:
While the empirical results are strong, the paper lacks theoretical guarantees about convergence, bias-variance trade-offs, or the conditions under which the bootstrapping approximation remains accurate.
In Figure 2(right), APPO with task and option rewards converges to a lower score than APPO with task reward, yet the paper did not try to analyze this phenomenon.

Presentation
Strengths:
This paper effectively articulates why scaling hierarchical RL is challenging, using concrete examples and figures to illustrate the parallelization difficulties. This sets up the contribution well.
This paper has a logical flow that is easy to follow, most of the technical concepts are clarified in a straightforward manner.
The supplementary material provides extensive details on architectures, hyperparameters, environment descriptions, and algorithmic pseudocode, demonstrating commitment to reproducibility.
Weaknesses:
Although it could be inferred from text, this paper did not explicitly show whether it is Cython that made this acceleration possible, or is it the system design that did the trick, which could be confusing for some.

Significance
Strengths:
Hierarchical RL has long promised benefits for complex tasks, but its computational inefficiency has prevented deployment at scale. SOL removes this barrier, enabling HRL to be applied to domains requiring billions of samples.
The throughput improvements (35-580x) are substantial enough that researchers working on other hierarchical methods could adopt SOL's design principles. The code release and clear engineering guidelines lower the barrier for adoption.
Weaknesses:
The focus is on throughput and asymptotic performance, not sample efficiency. This limits applicability to domains where interaction is expensive, even if compute is abundant. The performance improvement under environments other than ones mentioned in this paper could be less than expected.

Originality
Strengths:
SOL creatively combines existing concepts(i.e. options framework, bootstrapping, V-trace, high-throughput RL libraries) into a coherent system. The originality lies in identifying and resolving the specific friction points when scaling HRL.
The parallelized V-trace computation and the environment wrapper design are original engineering contributions that solve non-trivial parallelization challenges.
Weaknesses:
Theoretical analysis in this paper is moderate, yet not as comprehensive as existing works.

---

> ### Author Rebuttal · Authors · 2026-03-30
>
> Thank you for your time spent reviewing our paper and for the overall positive review. Please see our responses to your comments and questions below.
>
> > While the empirical results are strong, the paper lacks theoretical guarantees  [...]
>
> The focus of this paper is on designing a scalable and empirically effective HRL algorithm. However, although flat policy gradient algorithms are known to converge to local optima, such an analysis becomes more difficult in our setting because of the dependence between the controller and option policies. Note that prior analyses such as [1,2] do not apply in our case, because they optimize a single reward, whereas in our case, the option policies and controller optimize different rewards. We believe analyzing this type of algorithm would be interesting future work.
>
>
> > In Figure 2(right), APPO with task and option rewards converges to a lower score than APPO with task reward, yet the paper did not try to analyze this phenomenon.
>
> Adding the option rewards (Gold in this case) makes the agent adopt the suboptimal strategy of only gathering gold (+20 reward), whereas the agent using the task reward alone is able to do some local exploration and gather a little bit of gold (+3 reward) before heading to the staircase (+20 reward). We will add a note discussing this in the updated paper.
>
> > Although it could be inferred from text, this paper did not explicitly show whether it is Cython that made this acceleration possible, or is it the system design that did the trick, which could be confusing for some.
>
> We will clarify in the updated paper that the accelerations do not primarily come from the Cython code (which increases speed by around ~20%). We keep this brief due to rebuttal space constraints, but at a high level, the Cython code only becomes useful when we allow short option execution lengths, because they increase the number of controller reward sums that need to be computed, and this computation is difficult to tensorize because the execution lengths can vary in size.
>
> > The focus is on throughput and asymptotic performance, not sample efficiency. This limits applicability to domains where interaction is expensive, even if compute is abundant. [...] How is the compatibility of your work with other environments than gym Robotics? Could it also achieve a better throughput performance?
>
> It is correct that the target use case is high-throughput environments. That being said, our algorithm inherits PPO's ability to do multiple off-policy updates on the same batch of data, which can sometimes improve sample efficiency and maintain GPU utilization when the environment throughput is lower. To illustrate this we have run a new ablation experiment where we tune the number of PPO epochs for both MiniHack tasks in {1, 2, 4, 8}. On ZombieHorde, using 8 epochs results in a ~4x improvement in sample efficiency (with similar wall clock time) compared to the default setup, which uses 1 epoch. On TreasureDash the sample complexity is similar for the different numbers of epochs. This shows that on some tasks, the sample complexity can be reduced by increasing the number of epochs. We will add this to the updated paper. Also, in our Mujoco experiments we use 10 epochs, which was the PPO default in Sample Factory. In short, SOL can be run in any setting where PPO can be run.
>
> > It's noted in the paper that "efficient parallelized masking when computing the advantages and value targets for each policy". Is there a particular reason that you mentioned the concept "efficient masking"?
>
> In a standard flat agent, we only need to compute the advantages and value targets for a single policy. However, in our case, we are computing them for multiple different policies, which are mixed together in the same batch. Our process for doing this operates in a single FOR loop over the time dimension, using tensorized operations and masks, which is why we call it efficient. The full code for this operation can be found in Appendix D.4.
>
> > Have you tried to compare the scale of neural network between a flat and hierarchical architecture? Could the difference in performance be related with this factor?
>
> The hierarchical agent has only a small number of extra parameters: for the linear layer mapping the last hidden state to controller actions, and an embedding layer for the policy indices. This results in only 3% more parameters than the flat agent (4712981 vs. 4576523 parameters). We therefore do not think the performance difference comes from extra parameters. We will add a note clarifying this in the updated paper.
>
> We hope that our clarifications and paper updates address your comments, and that you will consider raising your score from Weak Accept to Accept. If you have remaining concerns that prevent you from doing so, please let us know and we will be happy to discuss further.
>
>
> [1] The Option Critic Architecture, Bacon et al. (AAAI 2017).
>
> [2] Learning Abstract Options, Riemer et al. (NeurIPS 2018).

---

> > ### Author Rebuttal · Reviewer_eAtn · 2026-04-02
> >
> > The authors have addressed the questions I raised, I will raise soundness score to 4, with overall recommendation from Weak Accept to Accept.

---

### Official Review · Reviewer_4Eqt · 2026-03-11

**Soundness:** 4
**Presentation:** 3
**Significance:** 4
**Originality:** 3
**Overall Recommendation:** 5
**Confidence:** 3

**Summary:**

The paper proposes a scalable hierarchical architecture, Scalable Option Learning (SOL), that achieves high throughput, enabling learning from billions of interactions. The paper demonstrates that, with hand-designed cumulants, SOL learns options that enable solving difficult tasks such as NetHack without any *special* phases, in settings where flat agents fail to solve the task. The paper further shows how SOL can be paired with methods to discover these cumulants, enabling its use with fewer biases/priors and at a larger scale. Finally, SOL is evaluated on MiniHack and MuJoCo, showcasing its general applicability.

**Compliance With Llm Reviewing Policy:**

Affirmed.

**Final Justification:**

The paper is well written and is very thorough in the experimental setup. The rebuttal addressed all of (minor) concerns, reinforcing my assessment.

**Key Questions For Authors:**

None

**Limitations:**

Yes

**Strengths And Weaknesses:**

**Strengths**
- The paper is well-motivated and well-positioned. The need for scalable RL architectures, given access to high-quality simulators, is highly relevant to current research directions.
- The experimental section is clear and detailed, explaining the choice of baselines and experiments. Furthermore, the experimental results are strong.
- The proposed method can be paired with any option discovery method, which could enable scaling many such existing methods.

**Weakness**
- The paragraph titled *Learner Worker* in Section 3.3 is difficult to parse, primarily due to an excessive concentration of information. The paper would greatly benefit from a rewriting of this paragraph.

**Minor Comments**
 1. Line 39 reads: "an objective which bootstraps option value functions upon option switches, allowing option policies to optimize independent rewards despite being trained with mixed behavior policies." Though not stated as a contribution, this reads as one. It is suggested to rephrase the paragraph to make explicit that this is a component of SOL, rather than a novel standalone contribution.
2. Line 111: "we approximate this expectation with a single bootstrapped Monte Carlo rollout" — The use of "bootstrapped" and "Monte Carlo" together feels *wrong*. A more precise term such as K-step return is more apt.
3. Figure 1 caption: "...and routs the..." appears to be a typo.

---

> ### Author Rebuttal · Authors · 2026-03-30
>
> Thank you for the time spent reviewing our paper and for the positive review. Please see our answers to your questions and comments below.
>
>
> > The paragraph titled Learner Worker in Section 3.3 is difficult to parse, primarily due to an excessive concentration of information. The paper would greatly benefit from a rewriting of this paragraph.
>
> We agree that this paragraph has a lot of information. If the paper is accepted, we will use the extra space allowed for the camera ready to rewrite it in a less dense manner that is easier to understand. In particular, we plan to do this by: i) adding pseudocode for the learner worker operations, which should be easier to understand than a text description, and ii) adding a figure illustrating the parallelized return and advantage computations, and how the masking and caching works.
>
>
>
> > Line 39 reads: "an objective which bootstraps option value functions upon option switches, allowing option policies to optimize independent rewards despite being trained with mixed behavior policies." Though not stated as a contribution, this reads as one. It is suggested to rephrase the paragraph to make explicit that this is a component of SOL, rather than a novel standalone contribution.
>
> We are happy to replace “SOL introduces” by “SOL includes” on Line 39.
>
>
> > Line 111: "we approximate this expectation with a single bootstrapped Monte Carlo rollout" — The use of "bootstrapped" and "Monte Carlo" together feels wrong. A more precise term such as K-step return is more apt.
>
> Thank you for the suggestion. We will use this terminology in the updated paper.
>
>
> > Figure 1 caption: "...and routs the..." appears to be a typo.
>
> Good catch - we will replace “routs” with “routes”.
>
>
> We hope that our clarifications and paper updates address your questions and comments. If you have any remaining questions, we will be happy to discuss further.

---

> > ### Author Rebuttal · Reviewer_4Eqt · 2026-03-31
> >
> > The authors have addressed the minor points I raised. I maintain my score.

---

### Decision · Program_Chairs · 2026-04-30

**Decision:**

Accept (spotlight)

**Comment:**

This paper aims to scale up hierarchical reinforcement learning in large-scale settings. The authors propose Scalable Option Learning (SOL), an HRL method specifically designed for training option-based RL in high-throughput environments. Compared to traditional HRL methods that use multiple option policies, SOL uses a single shared network to train all option policies, allowing efficient GPU batching. Experiments show that SOL achieves a ~35x speedup.

All reviewers agree that this paper is well-motivated and well-written. It discusses an important issue in scaling up HRL training, and SOL provides not only a practical system-level solution but also a solid empirical contribution. The experiments are comprehensive and convincing, including multiple environments and demonstrating the advantages of SOL. Although this paper is largely engineering, the authors did a great job demonstrating the effectiveness of their approach. Overall, I agree with the reviewers and strongly recommend acceptance, potentially as a spotlight if possible, as I believe scaling up HRL would be of interest to the RL community.